# Structural basis of voltage-dependent gating in BK channels

Gustavo F. Contreras [1], Rong Shen[1], Ramon Latorre [2] & Eduardo Perozo [1] ✉

The allosteric communication between the pore domain, voltage sensors, and $Ca^{2+}$ binding sites in the calcium- and voltage-activated $K^+$ channel (BK) underlies its physiological role as the preeminent signal integrator in excitable systems. BK displays shallow voltage sensitivity with very fast gating charge kinetics, yet little is known about the molecular underpinnings of this distinctive behavior. Here, we explore the mechanistic basis of coupling between voltage-sensing domains (VSDs) and calcium sensors in *Aplysia* BK by locking the VSDs in their activated (R196Q and R199Q) and resting (R202Q) states, with or without calcium. Cryo-EM structures of these mutants reveal unique tilts at the S4 C-terminal end, together with large side-chain rotameric excursions of the gating charges. Notably, the VSD resting structure (R202Q) also revealed BK in its elusive, fully closed state, highlighting the reciprocal relation between calcium and voltage sensors. These structures provide a plausible path where voltage and $Ca^{2+}$ binding couple energetically and define the conformation of the pore domain and, thus, BK's full functional range.

Physiological processes rely on feedback loops to maintain homeostasis under fluctuating metabolic conditions. The large-conductance calcium- and voltage-activated $K^+$ channel (BK) plays a fundamental role in establishing negative feedback loops to control the firing patterns of neurons, transmitter release, insulin secretion, and smooth muscle contraction[1]. BK's unique feature lies in its ability to integrate membrane potential and intracellular calcium levels while deploying a large $K^+$ conductance to control cell excitability efficiently. BK channels comprise four identical α subunits (Slo1)[2,3] with seven transmembrane segments (TMD) and a large C-terminal cytosolic domain (CTD). A voltage sensing domain (VSD, segments S0–S4) and a pore-gate domain (segments S5–S6) reside in the membrane. Additionally, two homologous regulators of the conductance of $K^+$ domains (RCK) in each subunit (RCK1 and RCK2) assemble as a tetrameric gating ring (4× CTD) and serve as the cytoplasmic $Ca^{2+}$ sensor[4].

BK structure is highly correlated to its function in response to chemical stimuli[5–7]. In BK, the closed-to-open transition (C→O) is cooperatively facilitated by intracellular $Ca^{2+}$ and membrane potential[8–10], although BK can open independently by either $Ca^{2+}$ binding[5,6,11–13] or depolarizing voltage[9,14–17]. BK can also open, albeit at very low probabilities, even when all voltage sensors are at rest and in the absence of

$Ca^{2+}$ (refs. 14,15). An allosteric model developed by the Aldrich[15,18,19] and Magleby[20,21] groups defines a theoretical framework to explain voltage- and $Ca^{2+}$ activation in BK channels. The Horrigan-Aldrich (HA) model[9,15], based on the modular tetrameric structure of BK, stipulates three modules: voltage sensors, pore domain, and $Ca^{2+}$-sensing domains. These modules are coupled through allosteric interactions represented by coupling factors. Since the modules are independent, BK can be activated by either voltage or $Ca^{2+}$ (refs. 14,15), which leads to simultaneous perturbations in the dynamics and equilibrium when individual sensors are active. Therefore, the expansion of the gating ring in response to $Ca^{2+}$ binding leads to conformational changes across all domains of the channel[5,6,11–13]. Voltage activation influences $Ca^{2+}$ binding to the RCK1 domain[22], triggering conformational rearrangements in the $Ca^{2+}$ sensor[23,24]. Reciprocally, $Ca^{2+}$ activation modulates the movements of VSD[25] and shifts the equilibrium constant that defines its resting-active transition[26]. The strength of the VSD-pore interaction and its relevance to activation remains under debate[15,26–28].

Given BK's dual activation regime, evaluating the major close-open state transitions should be possible by simply varying $Ca^{2+}$ at a constant voltage. Indeed, various BK cryo-EM structures have been obtained under apo and $Ca^{2+}$-bound conditions from a diversity of

[1]Department of Biochemistry and Molecular Biology, The University of Chicago, Chicago, IL, USA. [2]Centro Interdisciplinario de Neurociencia, Instituto de Neurociencia, Facultad de Ciencias, Universidad de Valparaíso, Valparaíso, Chile. ✉e-mail: eperozo@uchicago.edu

species[5-7,29-31]. As expected, structures of Ca²⁺-bound BK at 0 mV satisfy the expectations of a fully open state, yet analyses of Ca²⁺-free (closed) structures have been far less clear. This uncertainty stems from the fact that in the nominal absence of Ca²⁺, the intracellular gate remains sufficiently wide to allow a hydrated K⁺ ion or even large quaternary ammonium ions to pass through or access the inner water-filled cavity[32], casting doubt on whether these structures truly represent closed conformations[33]. The HA allosteric model predicts that, at 0 mV without Ca²⁺, the channel may adopt intermediate closed states, where at least one voltage sensor is active[9,15]. Further complicating the mechanistic interpretation of the apo structures, large hydrophobic ions can to block the channel in putatively closed states, yet fully closed BK, with all VSDs at rest, is unaffected by these blockers[34]. This observation also raises questions regarding the VSD conformation in the apo structure, which reveals only minor S4 movements, with no apparent rearrangements in the interaction between gating charges and their countercharges. The number of BK gating charges remains under scrutiny, which adds to this uncertainty. A decentralized voltage sensor was proposed to account for BK's uniquely shallow voltage dependence and fast gating kinetics[35], where the gating charges are spread among S2, S3, and S4. However, gating current measurements of charge mutants show[36], that only residues R210 and R213 contributed to the overall gating charge.

Determining the extent to which BK's VSD activation influences other channel domains is fundamentally important for establishing the mechanism of BK signal integration. Here, by using electrostatic engineering of the S4 gating charges, we have established the mechanistic basis of the bidirectional coupling between voltage-sensing domains (VSDs) and calcium sensors in *Aplysia* BK. We utilize single-particle cryo-EM, molecular dynamics simulations, and electrophysiological approaches to reveal the high-resolution structures of *Aplysia* BK under conditions that demonstrate its cooperative gating. We show that the nature and kinetics of VSD function in BK can be fully explained by a mechanism in which limited S4 helical excursions and large rotameric reorientations in the gating arginines satisfy charge movement, helping to explain its shallow voltage sensitivity and fast kinetics. Simultaneously, determining a much sought-after closed BK structure provides insights into the concurrent coupling between the VSD and Ca²⁺ sensing domains. These structures, and those obtained with and without bound Ca²⁺, serve as foundational frameworks for understanding BK allosteric interactions and their role as signal integrators in excitable cells.

## Results

### Neutralization of S4 charges polarizes the voltage sensor structure

To evaluate their structural role in channel activation, we systematically neutralized, one at a time, the charged residues of the S4, R196 (R1), R199 (R2), and R202 (R3) in the *Aplysia* BK (aBK) channel (Fig. 1a), based on the construct previously used for the structural determination of the *wild-type* (WT) structure of *aBK*[5,6]. In virus-infected sf9 cells, aBK VSD mutants generate large shifts in the conductance-voltage relationship (G-V), similar to those found in mouse and human BK[35-37]. The half-activation voltages ($V_0$) of R1Q (6.45 ± 2.94 mV) are shifted leftward relative to the WT (173.4 ± 2.64 mV), while R2Q shifts rightward (228.97 ± 10.52 mV) with a decrease in $z$ (0.35). No macroscopic current was detected for the R3Q mutant (Fig. 1b). These $V_0$ shifts support a conserved voltage activation mechanism among different BK channel species.

The structures of these aBK channels with neutralized voltage-sensing arginines were determined under Ca²⁺-bound and apo conditions in detergent micelles using single-particle cryo-electron microscopy (cryo-EM). These correspond to residues R207, R210, and R213 in the human BK channel. Ca²⁺-bound mutants were purified in solutions containing 10 mM Ca²⁺ and 10 mM Mg²⁺, whereas the apo condition was purified with 2 mM EDTA. Except for R3Q-apo, all data sets

produced a single major density with C4 symmetry. Although refinements with C1 and C2 symmetries were tested, but resulted in models with lower resolution. The R3Q-apo data set, however, led to two classes with C4 symmetry. The resolutions attained for each mutant were as follows: for R1Q, 2.7 Å in apo and 2.9 Å in Ca²⁺-bound conditions (Fig. S1a, b); for R2Q, 3.4 Å in apo and 2.9 Å in Ca²⁺-bound conditions (Fig. S2a, b); and for R3Q, 3.6 Å in apo and 2.6 Å in Ca²⁺-bound conditions (Fig. S3a, b and Table S1). Additionally, we solved the structure of the mutant F304A in Ca²⁺-bound condition with a resolution of 3.0 Å (Fig. S4a). For all mutants, whether Ca²⁺-bound or under apo conditions, comparative analysis of the refined atomic models revealed little or no changes in the positions of the α-carbons for the S4 arginines (Fig. 2a), displaying remarkable similarity with previously obtained VSD structures[5-7]. However, considerable rearrangements were observed in the rotameric conformation of the gating charge arginines, which we propose to shed light on the molecular mechanisms underlying BK voltage sensing.

### Structural implications of the activated voltage sensor

As shown previously in hBK[36,38] and mBK[35], the activated state of the VSD can be stabilized at 0 mV by neutralizing the first arginine in the S4 segment of BK. We found that R1Q decreased the activation energy of the VSD, as evidenced by a ~ 400 mV leftward shift in the Q-V curve toward negative potentials (Fig. 2b), resulting in channels with facilitated activation, without changes in voltage sensitivity or coupling between the VSD and the pore[35,36]. When compared with the WT Ca²⁺-bound structure, R1Q displays the same conformation in the TM region (segments S1–S6, Cα RMSD < 1 Å; Fig. S5a). In the R1Q structures, R2 establishes a salt bridge with D142 (S2) at approximately 2.7 Å, while R3 interacts with D175 (3.2 Å) and F149 (3.2 Å) (Figs. 2a and S5b, c). At the intracellular end of S4, the aromatic ring of Y212 interacts with the amide group of the D208 backbone in S4 and G94 in S0' (amide-π). The S2–S3 loop moves down by approximately ~3.3 Å at position N161 and ~2.6 Å at D162. R38 in S0 establishes a salt bridge with D162 (2.8 Å) in the S2–S3 loop, and the carbonyl group of N161 forms a hydrogen bond with K85 (2.8 Å) at the S0' segment (Fig. S5d). These interactions are consistent not only in R1Q (both apo- and in Ca²⁺-bound) but also in the WT Ca²⁺-bound state. The introduction of glutamine at position R1 leads to a more negative extracellular surface in the VSD, created by residues D120 and D142 (Fig. S6a, b). This local potential likely drives residues R2-R3 upward, stabilizing the active state of the VSD in the absence of Ca²⁺. This suggests that in the WT channel, R1 tends to drive the VSD toward its resting (down) conformation in the apo condition (Figs. 2a and S6a, b) and confirms that R1 does not act as a gating charge but instead appears to modulate VSD transitions. This result indicates that in R1Q, the VSD represents the activated (Up) conformation in the WT channel. The present R1Q apo structure displays further detail in the S0 helix (Fig. S7a, b), allowing us to resolve five additional residues at the N-terminal end of S0 (Fig. S7a, b). This higher structural order may be attributed to the non-canonical interaction between S4 and S0 (Fig. S7a,b), involving aromatic residues at the extracellular end of S0 coupling with S4 and S5 and the S0´s C terminal with S3 (Fig. S7b). These findings suggest that the S0 helix plays a functional role as part of the voltage sensor and is involved in determining the voltage activation mechanism.

As anticipated, voltage sensor activation opens the channel without calcium binding, as shown in the R1Q apo structure (Fig. 3a). Not only is the pore domain in an open configuration but also the Ca⁺² sensor appears expanded in the apo condition despite the absence of a strong density for Ca²⁺ at its binding site in the R1Q apo (Fig. S8a, b). In the WT channel, Ca⁺² binding triggers an expansion in the CTD, compared to the WT apo condition (Cα RMSD ~ 4 Å). We found minor differences between apo and Ca²⁺-bound conformations in the R1Q structures (Cα RMSD ~ 1.6 Å), suggesting that the active VSD drives the CTD structural rearrangement (Fig. S8a, b). This finding provides

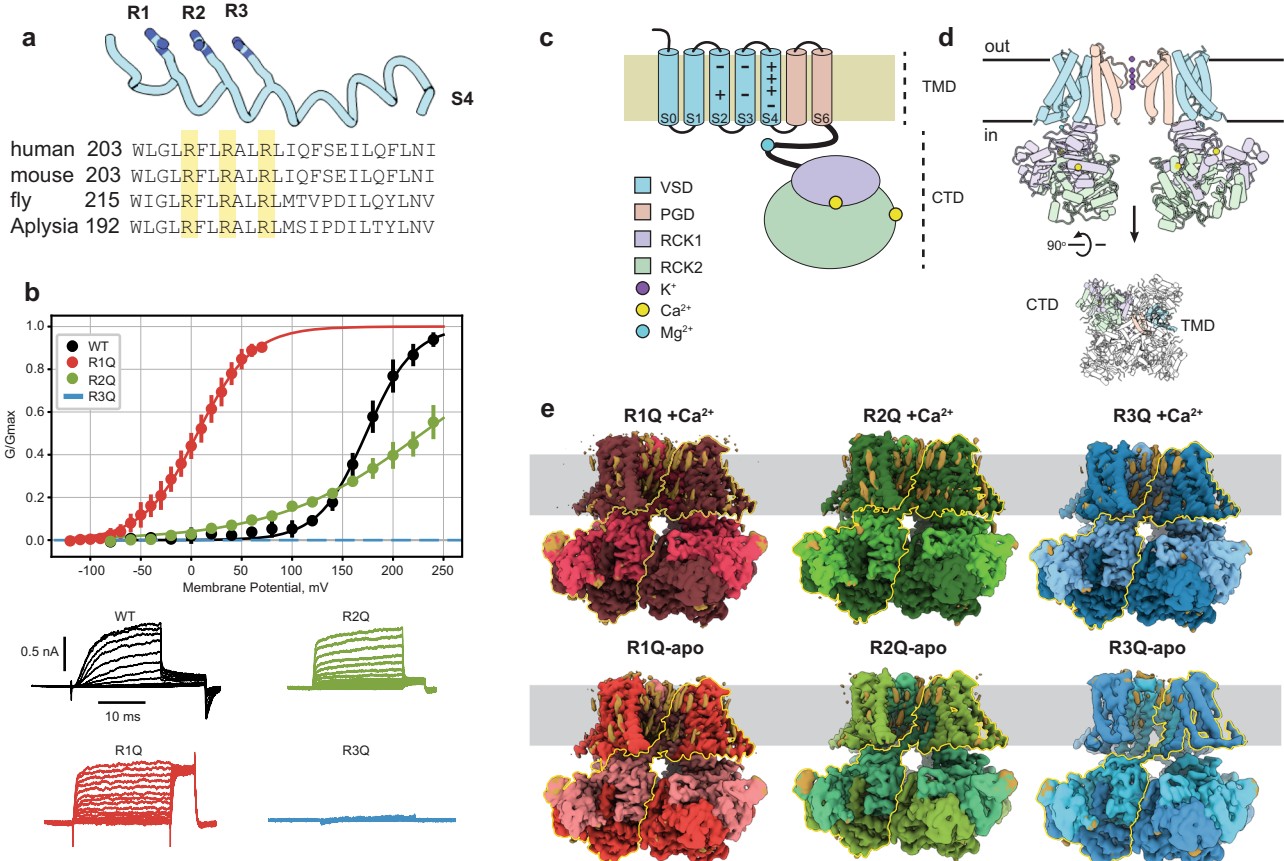

**Fig. 1 | Function and architecture of *Aplysia* BK. a** Sequence alignment of BK S4 across different species. Labels R1, R2, and R3 designate the mammalian arginine residues that alter the state of the VSD when calcium is absent. **b** G-V relationships and representative traces of WT, R1Q, R2Q and R3Q. The fitted parameters to a Boltzmann distribution were: WT ($z = 1.10 \pm 0.08$, $V_0 = 173.60 \pm 2.64$ mV, $n = 5$), R1Q ($z = 0.96 \pm 0.05$, $V_0 = 6.31 \pm 2.94$ mV, $n = 6$), and R2Q ($z = 0.35 \pm 0.03$, $V_0 = 231.44 \pm 10.52$ mV, $n = 4$). Data are shown as mean ± SEM. K⁺ currents were recorded in sf9 infected cells under a voltage clamp in whole-cell configuration; the internal solution contained 10 mM EGTA. **c** Schematic of aBK secondary structure features. **d** Side and top views of the aBK structure are colored according to different domains. For clarity, we show a side view with only two subunits. **e** Cryo-EM maps of aBK mutants in apo- and Ca²⁺-bound, colored opaque for VSD and RCK2, and light for the pore domain and RCK1. The figures maintain the same color scheme throughout.

direct structural evidence of a robust coupling between the VSD and CTD underlying channel´s gating.

Mutant R2Q also led to active-like VSD structures, regardless of Ca²⁺ binding (Figs. 2a and S8a, b). However, R2Q VSD lacks direct coupling between S4 and S2 at residue R2-D142 as seen in the active VSD structures of R2Q and WT (Figs. 2a and S5b). This coupling has been tracked using fluorescent probes attached to the extracellular end of S4 and S2[39]. Neutralization of either R2 or D142 resulted in the loss of a voltage-dependent S4-S2 coupling and reflected an increase in the activation energy to open the channel[39]. This result suggests that the interaction between S2 and S4 plays a pivotal role in voltage sensitivity and the VSD-pore coupling.

**Structural Implications of the Resting Voltage Sensor**
Neutralizing arginine at position R3 (R202Q) drives the VSD towards its resting state, as suggested by the rightward shift of Q-V in the homologous mutant in hBK[36]. The remaining charged residues in S4 (R1 and R2) maintain similar positions in the R3Q-apo and R3Q Ca²⁺-bound structures. R1 forms a salt bridge with D120 (S1) with a bond length of 2.7 Å, while R2 couples with D175 (S3) and exhibits weak interactions with F149 (S2) (Fig. 2a and Fig. S5b, c). These two interactions contribute differently to channel gating. The interaction at the extracellular side of the VSD between R1 and D120 has a modulatory role. The zone flanked by D175 (S3) and the aromatic ring F149 (S2) appears to play a role in coordinating charge transition during VSD activation,

coordinating R2 in the resting (down) conformation and R3 in the active state, reminiscent of the hydrophobic plugin in Kv channels (Figs. 2a and S5b, c). Residue Y212 (S4) is packed between the backbone nitrogen of D208 and R100 (S1) between S4 and S1 (Fig. S5c), where its importance has been highlighted by the effect that alanine substitution at hF223A (aY212). This mutation alters voltage sensitivity by increasing channel open probability at negative potentials while decreasing it at positive voltages. Notably, BK cannot reach maximal activation without calcium, indicating that the interaction between S4 and S0′ may be critical for channel opening[28]. Neutralization of the primary counturcharge in the resting VSD, hD186N (D175), reduces the coupling between VSD activation and channel opening, as previously described[35,36]. The resting VSD structure (R3Q-apo) shows that R2 undergoes conformational changes during VSD activation. In the resting VSD structure (R3Q-apo), R2 faces the intracellular solution. In contrast, the active VSD state (R1Q-apo) faces the extracellular solution (Fig. 2a, b). R2 forms obligatory interactions in both casesbecause neutralizing its state-dependent counturcharge uncouples the VSD from the pore and dramatically affects its voltage sensitivity. This finding agrees with the functional characterization of residue R2 and confirms that R2 is the main gating particle of the BK VSD[36].

We suggest that the R3Q-apo (resting VSD) and R1Q (active VSD) structures define the conformational limits of the functional transitions in the BK voltage sensor. Therefore, the activation of the VSD mostly involves an upward change (towards the extracellular face of

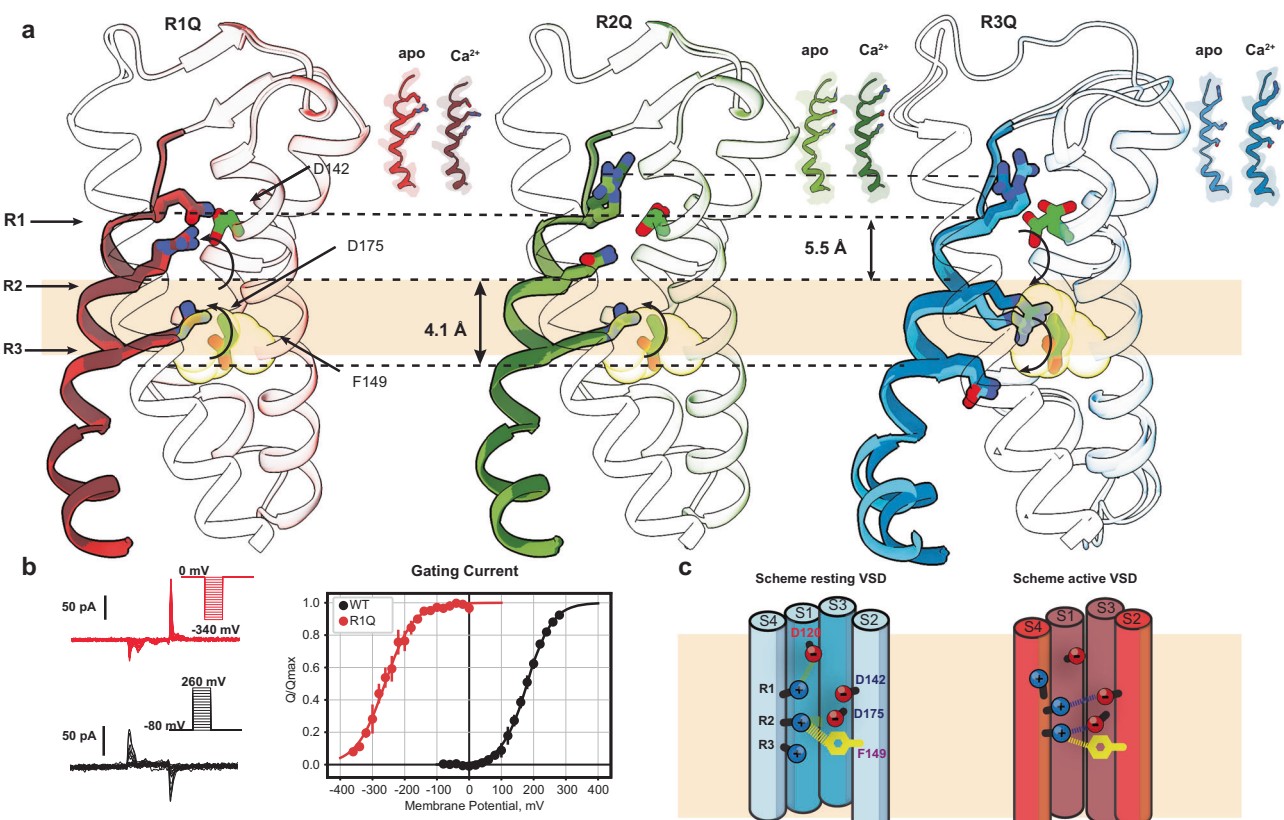

**Fig. 2 | S4 charge neutralization stabilizes distinct VSD conformations independent of Calcium. a** The VSD S4 of aBK mutants R1Q (red), R2Q (green), and R3Q (blue) are shown as a cartoon superimposed for $Ca^{2+}$-bound (opaque) and apo (light). To ensure clarity, we depict only S4 as a cartoon, while TMD S1, S2, and S3 remain transparent. Stick side chains are shown for the charged residues 196,199, and 202. The structures were aligned based on the residues of their selectivity filter 275–282 (S5–S6 loop, the least mobile part of all). The speculatively hydrophobic gasket is focused on the electric field marked by the transparent khaki slab.
**b** Representative traces of gating current and steady-state Q-V curves for mutants R1Q (red) and WT (black). Q-V curves were obtained by 300 μS integration of the

Qoff response. The half-activation voltage was determined by fitting the data to a Boltzmann distribution, yielding $V_0 = 179.82 \pm 8.6$ mV and $z = 0.6500 \pm 0.130$ for WT ($n = 4$), and $V_0 = -231.33 \pm 1.78$ mV and $z = 0.5761 \pm 0.0216$ for R1Q ($n = 5$). Data are shown as mean ± SEM. The apo condition was achieved with 5 mM EGTA.
**c** Schematic representation showing the resting and active VSD conformational states. In the resting VSD scheme, green dotted lines indicate hydrogen bonds with a distance of <3.5 Å. In the active VSD scheme, these interactions are represented by violet dotted lines. The yellow dotted lines in both schemes indicate possible π-cation interactions with distance <4 Å.

the channel) in the R2 rotamer orientation of approximately 6 Å while R3 reorients some 5 Å (Fig. 2a and Movie S1). This reorientation is coordinated by the counter charge D175 at S3 and D142 at S2, together with state-dependent interactions between R2 and D175 and D142, leading to the coupling between the peripheral segments of the VSD (Fig. S5b, d and Movie S2). Those interactions generate explicit conformational changes at the bottom part of the S4 helix, which rotates from the M204 α-C by about 10°. Meanwhile, S0' turns about 8° toward the cytoplasmic side, causing a change in the packing of the aromatic ring Y212. At 0 $Ca^{2+}$, residue Y212 (S4) is packed between the backbone nitrogen of I209 and R100 (S1) between S4 and S1 (Fig. S5c and Movie S3).

**The resting VSD structure reveals a deep closed state of the pore**
Remarkably, the VSD structure of apo R3Q is like apo WT but results in a considerably narrower pore (radius ~4.2 Å, HOLE) compared to the WT apo structure (radius ~5.7 Å, HOLE) (Fig. 3b). This narrowing exposes a more hydrophobic profile due to the side-chain rotations of residues F304 and I308 (Fig. S9a and movie S4). Although both classes of R3Q-apo have narrower pores than the WT apo structure, we chose the class with the narrowest internal vestibule for structural comparison (Fig. S3a). In this structure, the side chain of I308 represents the narrowest point in the permeation pathway (radius ~4.2 Å, HOLE) (Fig. 3b, c). A hydrophobic pore with a diameter <10 Å is expected to

undergo de-wetting[40,41], posing a significant free energy barrier to ion conduction and impeding the permeation of a hydrated $K^+$ (radii 3.5–4.0 Å). This structure represents the narrowest internal vestibule of a BK channel.

We suggest that R3Q stabilizes a deep closed state of the BK pore and that conformation is evidence of an activation-deactivation gating mechanism based on reorienting the S6-based inner bundle helix. This mechanism appears to be conserved in other RCK-containing $Ca^{2+}$-dependent K+ channels as recently suggested for MthK[42]. Indeed, previous MD simulations[43–45] support the idea that the barrier for ion permeation in the closed state results from de-wetting within the constraints of a narrow hydrophobic pathway. Similar mechanisms have been proposed for other channels[46–48]. This deep closed state stabilized by mutant R3Q apo appears to be equivalent to the fully closed state of BK in the presence of resting potential. The introduction of Glutamine at R3 likely forces the voltage sensor into its resting state, as it cannot occupy the hydrophobic center. Furthermore, early functional studies showed that at saturating $Ca^{2+}$ concentrations, the Po at 0 mV following charge neutralization at R3 was less than $10^{-2}$ (ref. 35). The gating current from a homologous mutation in hBK also shows shallow voltage dependence for R3Q and a decrease in coupling to the pore[36]. Surprisingly, R3Q channels managed to transition to an open state under $Ca^{2+}$-bound conditions. This indicates that the conformational changes in the gating ring must affect the structure of the TMD.

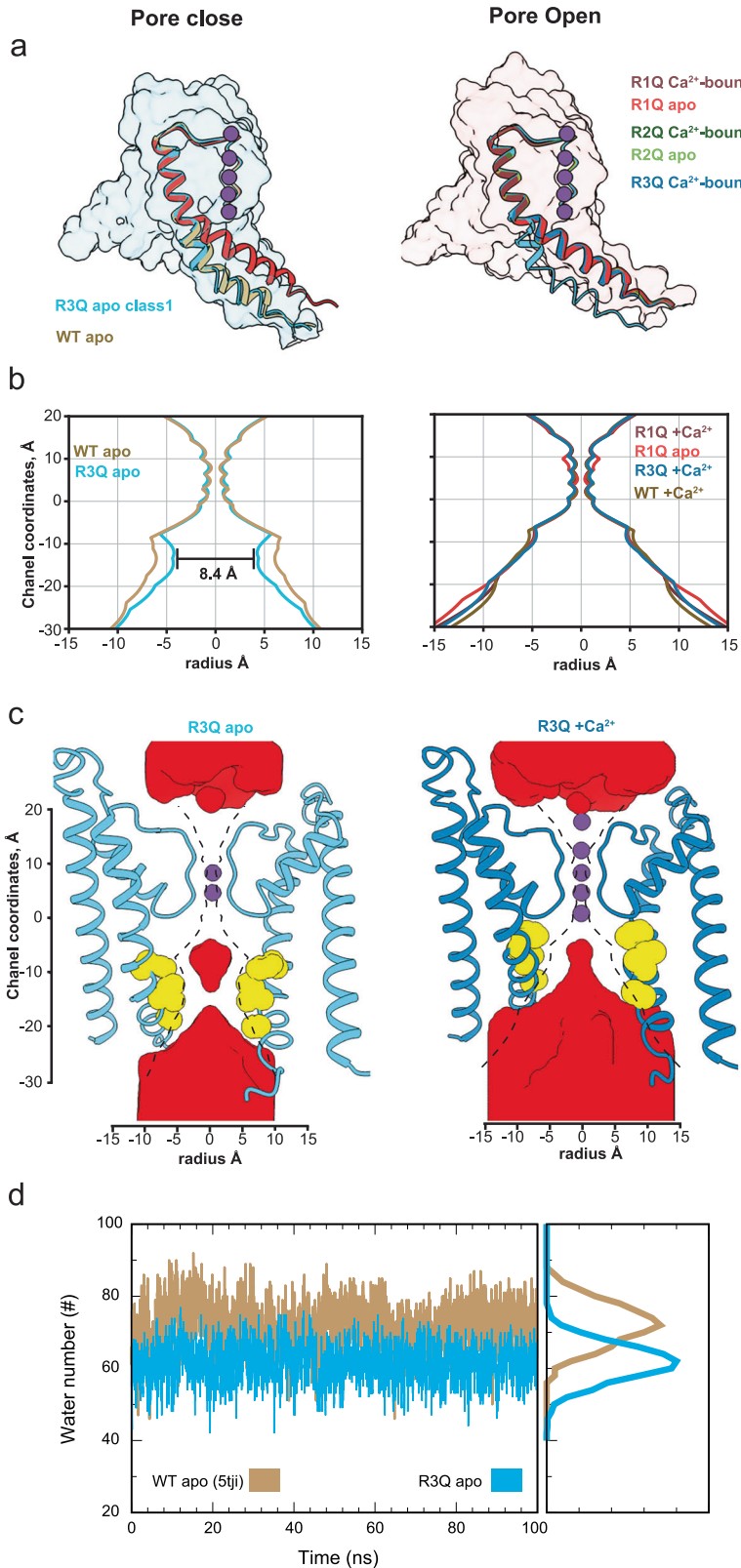

To determine whether R3Q is in a closer state of the pore than the WT apo structures, we performed MD simulations of the pore domain using the R3Q closed state and WT apo structures while restraining the positions of the α-carbons in the S6 helix. These simulations showed that in the R3Q apo structure, fewer water molecules are stabilized in the internal cavity (Fig. 3d). When the structural constraint is removed, the cavity becomes devoid of water,

and lipids start to populate the inner cavity using lateral windows. Rotation of the S6 helix is required to close the channel, as it exposes its hydrophobic surface to the conduction pathway and opens membrane-facing fenestrations in the closed structures (Fig. S9c). These fenestrations also appear in other BK structures in apo condition but not in Ca$^{+2}$-bound structures[5–7,29,30]. They are likely alternative pathways that allow hydrophobic molecules to reach the inner

**Fig. 3 | A deep close state revealed by R3Q-apo. a** Structural comparison of pore-lining segment S6. We show the TMD of monomers R3Q-apo (light blue) and R1Q-apo (red) as surfaces that represent the pores in their respective closed and open conformations. The bending at S6 is illustrated by the atomic model in the cartoon images of the closed (R3Q-apo class 1 and WT-apo; PDB 5TJI, shown in light blue and tan) and open (R1Q-Ca$^{2+}$-bound, R1Q-apo, R2Q-Ca$^{2+}$-bound, R2Q-apo, R3Q-Ca$^{2+}$-bound, and WT-Ca2 + -bound; PDB 5TJ6, shown in crimson, red, olive, light green, steel blue, and sandy brown) structures for the amino acids numbered 274 to 320. **b** Pore-size profiles of the permeation pathway, calculated by HOLE[92] with a resolution of 1 Å, for the closed (left) and open (right) structures. **c** Side view of the pore

of R3Q, with only two subunits shown for clarity. The effective opening radius was determined using HOLLOW[93] by filling the protein's holes and empty spaces with virtual overlapping oxygen atoms placed at regular distances in a 0.5 Å cubic grid. The molecular surfaces (red) were calculated for a 4 Å-radius probe (about the size of hydrated K$^+$). The HOLE calculation is indicated by the black dashed line for the closed (R3Q-apo) and open (R3Q-Ca$^{2+}$-bound) structures. **d** The number of water molecules in the pore over time from two example MD simulations of WT-apo (tan line) and R3Q-apo class 1 (light blue line) structures. See "Methods" for definitions of pore water molecules.

cavity in MthK[42] and dSlo1[30]. These membrane-exposed windows further augment the hydrophobicity at the internal cavity, increasing the apparent affinity for hydrophobic ions.

In BK, quaternary ammonium (QA) blockers may access the central cavity in the open and closed states of the channel[32], unlike other Kv channels[49]. However, bbTBA, a bulky derivative of tetrabutylammonium, does not block the channel in a complete state-independent fashion[34]. Functional evidence suggests that bbTBA blocks the channel in an intermediate close state, but the fully closed BK channels are not sensitive to block by bbTBA[34]. We tested whether the geometry of the pore and its lateral fenestrations allows the entry of bbTBA to the internal cavity. We found that in 0 Ca$^{2+}$, the permeation pathway observed in other BK structures[5–7,29–31] is large enough to allow bbTBA access to the blocking position (Fig. S10a, c). However, the tight closure at I308 was sufficient to occlude entry of bbTBA in R3Q apo (Fig. S10b, d). This observation suggests that previous BK apo structures are consistent with a partially closed state, and only when the VSD is in its deeper resting state (R3Q apo) does the channel finally adopt a fully closed conformation. We submit that this structure is conformationally equivalent to the closed BK channel at negative potentials.

## Coupled conformational changes between voltage and calcium sensor

Compared to the open Ca$^{2+}$-bound R3Q structure, the fully closed R3Q-apo shows small conformational changes in the CTD. However, relevant differences were found in the αB helix. In the R3Q-apo, the αB helix is partially unfolded at its ends (residues: L374-K381), unlike in the R3Q Ca$^{2+}$-bound state where the αB helix (residues: L372-R382) (Fig. S11a, b) is reminiscent of all our open structures. Introducing a proline into different positions of the αB helix reduces both voltage and Ca$^{2+}$ activation, suggesting that the structure of the αB helix is crucial for the interaction between the CTD and VSD. Specifically, the structure of the hL390P mutant shows the first two residues of its αB helix (hBK: $_{385}$LE$_{386}$) are unfolded, like in R3Q-apo strcuture[50]. The resemblance in the αB helix structure of hL390P and R3Q apo (Fig. S11a, c) suggests that folding of the initial residues (hBK: $_{385}$LE$_{386}$; aBK: $_{384}$LE$_{385}$) of αB helix is essential for efficient coupling of CTD-VSD (Fig. S11d).

To evaluate whether the conformational change in the CTD could modify the structure of the resting VSD, we overlaid the structures of the gating ring from R3Q Ca$^{2+}$-bound with the TMD of R3Q apo. We observed clashes (superposition) between the terminal residues of the αB helix (L374 and R382) and S4 (T211) and S5 (R223) of R3Q-apo (Fig. S12a). The αB helix of WT Ca$^{2+}$-bound (5TJ6) also superposes with S4 and S5 of the WT apo structure (5TJI; Fig. S12b). We found similar results when comparing the CTD of hBK Ca$^{2+}$-bound with the TMD of the hBK apo structure (6V3G; Fig. S12c). This observation indicates that the conformational change of the αB helix is pivoting the movement of the intracellular ends of S4 and S5 helix in BK. The movement of these helices is based on hinge points R223 and T211, which are pivoted by L374 and R382 at the αB helix. Thus, the open state observed in the R3Q Ca$^{2+}$-bound condition may be achieved through the interaction

between L374 and R223, which underlies the coupling between S5 and S6 observed in all BK open structures.

The structure of the R1Q apo also shows an active-like conformation for the Ca$^{2+}$ sensor in its apo state, this indicates that VSD activation likely impacts the conformation of the gating ring. To investigate this possibility, we superimposed the active VSD (R1Q apo) structure with the CTD of R3Q apo, revealing clashes between the S4 and αB helix, as well as between the S2–S3 loop and the C-linker (Fig. S13a, b). Specifically, T211 at S4 pivots the αB helix at position F385. Meanwhile, N161 at the S2–S3 loop and D85 at S0' facilitate the pivot of the C-linker at position E328. The mutational analysis identifies these residues at the interface between the VSD (hD99 and hN172) and the CTD (hE374 and hE399) as crucial in coordinating Mg$^{2+}$ (refs. [51–53]), affecting the coupling to the VSD[53–55]. At saturating Ca$^{2+}$ concentrations, alanine substitution at position T211 in mBK (mQ222) shifted activation by more than 50 mV in the positive direction and could not be measured in 0 Ca$^{2+}$. Although there are non-conserved features between the C-linker of Aplysia and human BK (Fig. S13a), overlaying the Ca$^{2+}$-bound hBK TMD with the CTD of the apo hBK structure also results in clashes between the S4 and αB helix and between the S2–S3 loop and the C-linker (Fig. S13c). These observations emphasize the interplay between VSD and calcium sensors through the conformational changes within the BK channel structures.

## Reciprocal interactions between voltage and calcium sensor

The CTD at 0 Ca$^{2+}$ exhibits a weak interaction with segment S5–S6 at PGD in the R3Q-apo structure (Fig. 4a). Particularly, the carboxyl side chain of E377 in the αB helix forms a hydrogen bond with the hydroxyl side chain of S219 (2.7 Å) in S5 (Fig. 4a). Additionally, van der Waals interactions between S5 and S6 (R223 and K317) from adjacent monomers are reminiscent of RCK1 αB (L374) contacting its neighboring S6 (K320). In the Ca$^{2+}$-bound R3Q structure, a network of interactions between RCK1 and VSD is established (Fig. 4b). This network connects the αB helix to S5 and S4, with non-covalent interactions involving residues L374-R223, R382-T211, and R382-N214. Additionally, hydrogen bonds connect the αB helix to S4 (K381-D208; 3.2 Å) and the C-linker to S0' (K331-D86; 3.2 Å). The αB-βC loop also interacts with S0' through residues F384-S93 and T385-E90. A similar cluster of residues connects the CDT with the VSD in the R1Q-apo structure (Fig. 4c). Specifically, K331 in the C-linker of the CDT interacts with D86 (3.2 Å) in S0'. This is the primary electrostatic interaction that coordinates the coupling between the CTD-VSD sensors (Fig. 4b). This finding is consistent with mutagenic studies showing that alanine substitution at this position (hD99A) prevents Mg$^{2+}$ binding[53] and suggests that the molecular determinants for the coupling between VSD and CDT in the R1Q apo structure are the same as those mediated by Mg$^{2+}$ binding. Under apo conditions, the distance between residues N161 and E388 seen in the R1Q (4.3 Å) and R2Q (4.7 Å) variants is equivalent to the distance in the WT Mg$^{2+}$ (4.3 Å between N161-E388) bound structure. These findings indicate that the primary pathway for channel opening resides in the protein's intracellular interface. Calcium binding couples the CTD αB helix with the VSD S4-S5 loop, whereas VSD activation and Mg$^{2+}$ binding couples the CTD C-linker

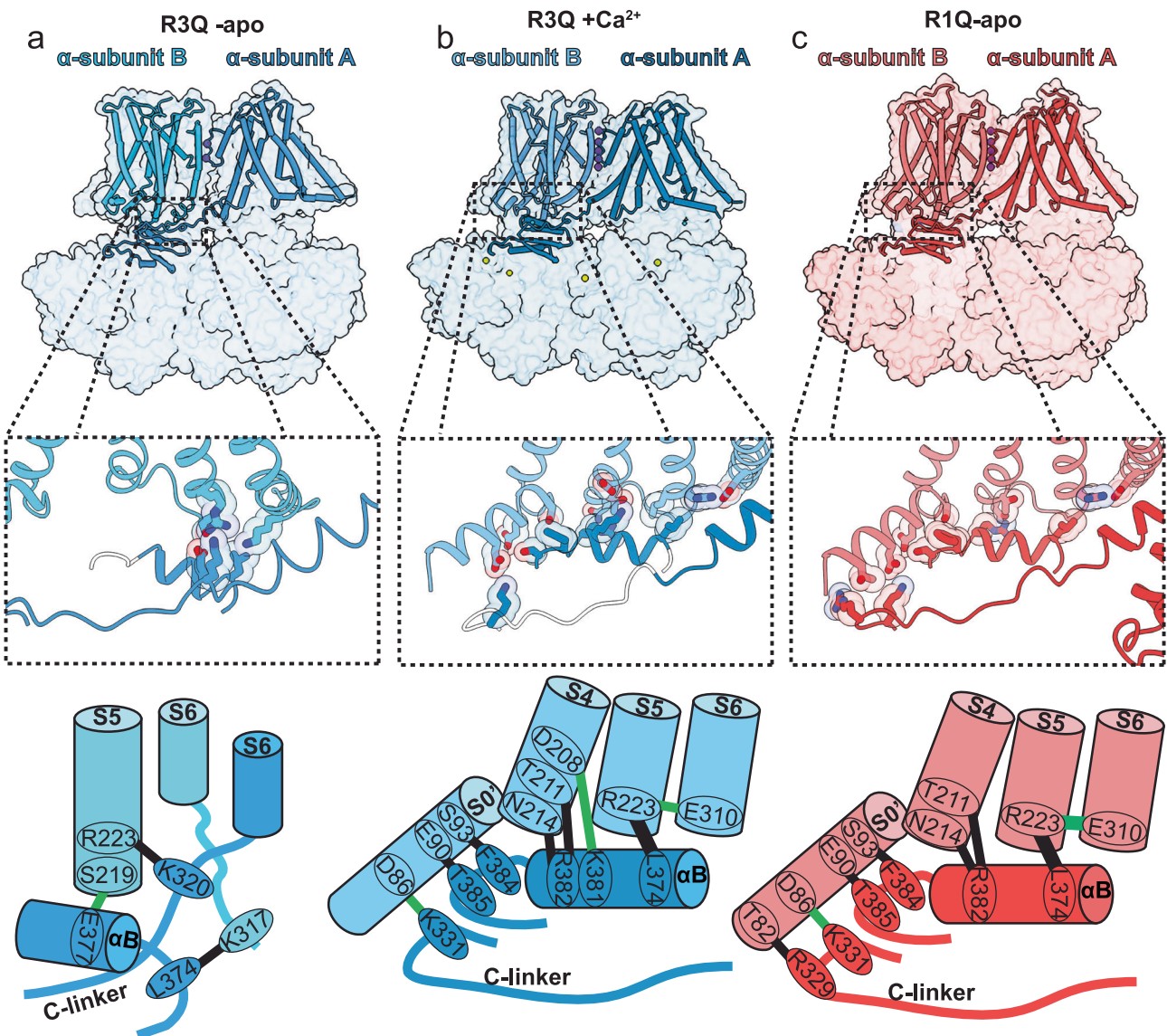

**Fig. 4 | Reciprocal interaction between VSD-CTD. a–c** (top) Composed view of two adjacent α-monomers. A cartoon represents the TMD and RCK1, while a transparent surface depicts the structure of the entire monomer overlay. (middle) zoom of the interaction surface between VSD-CTD; the key interacting residues are shown as stick representations superimposed with their transparent sphere representations. (bottom) scheme highlighting the key residues stabilizing the interaction between VSD-CTD.

with the VSD S0' helix. Thus, electrical and chemical stimuli reciprocally activate both sensors, utilizing a common interface to couple the sensors and leveraging similar energy pathways to open the channel.

The interaction between sensors leads to the formation of hydrogen bonds between residues R223 and E310, connecting S5 to the pore lining helix S6. Mutations in either R223 or E310 (hK234 and hE321) reduce the energy coupling to such an extent that it cannot be measured without calcium[28]. Despite the presence of this electrostatic interaction between S5 and S6 in all open channel structures, functional evidence points to a coupling between the VSD-CTD independent of the state of the pore. This coupling suggests that additional components might participate in the opening of the gate.

**Conformational changes in the pore during the closed to open transition**

Channel opening through either an activated VSD -apo condition or a resting VSD under Ca²⁺-bound conditions share similar structural elements. In the R1Q (apo and Ca²⁺-bound), R2Q (apo and Ca²⁺-bound), and R3Q Ca²⁺-bound structures the S6 gate adopts an open

conformation (Fig. 3a). S6, tilted at position M298-I299 (Fig. 5a) moves upward (6.7 Å at Cα 315), resembling the position of the Ca⁺²-bound WT structure (5TJ6; Fig. 3b). In the inner cavity, F304 forms a likely π-amide with the S275 backbone nitrogen (~4.3 Å; Fig. 5b), while F304 contacts I308 independent of the state of the pore. This pair of residues close the fenestrations in the open state and constricts the conduction pathway by exposing their side chains towards the inner cavity in R3Q-apo. Located next to the helix hinge bending point, I301 experiences a similar transition.

The importance of the hydrophobic interactions in BK S6 has been previously highlighted. Surprisingly, mutations at position I301 and F304 lead to opposite results. Most of the mutations at mL312 (I301) increase the open probability, and those that generate the largest G-V shifts in 0 Ca²⁺ (L312C, L312V, L312S, L312Q) also reduce Ca²⁺ sensitivity so that G-Vs in different Ca²⁺ concentrations superimpose[56]. Unlike mutations at mL312, many mF315 mutations (F315A, F315G, F315I, F315L, F315T) lead to undetected currents under physiological conditions[56]. Alanine substitution at hF315 (aF304) generates stable closed channels[38], even at saturating Ca²⁺ concentrations[27]. This result

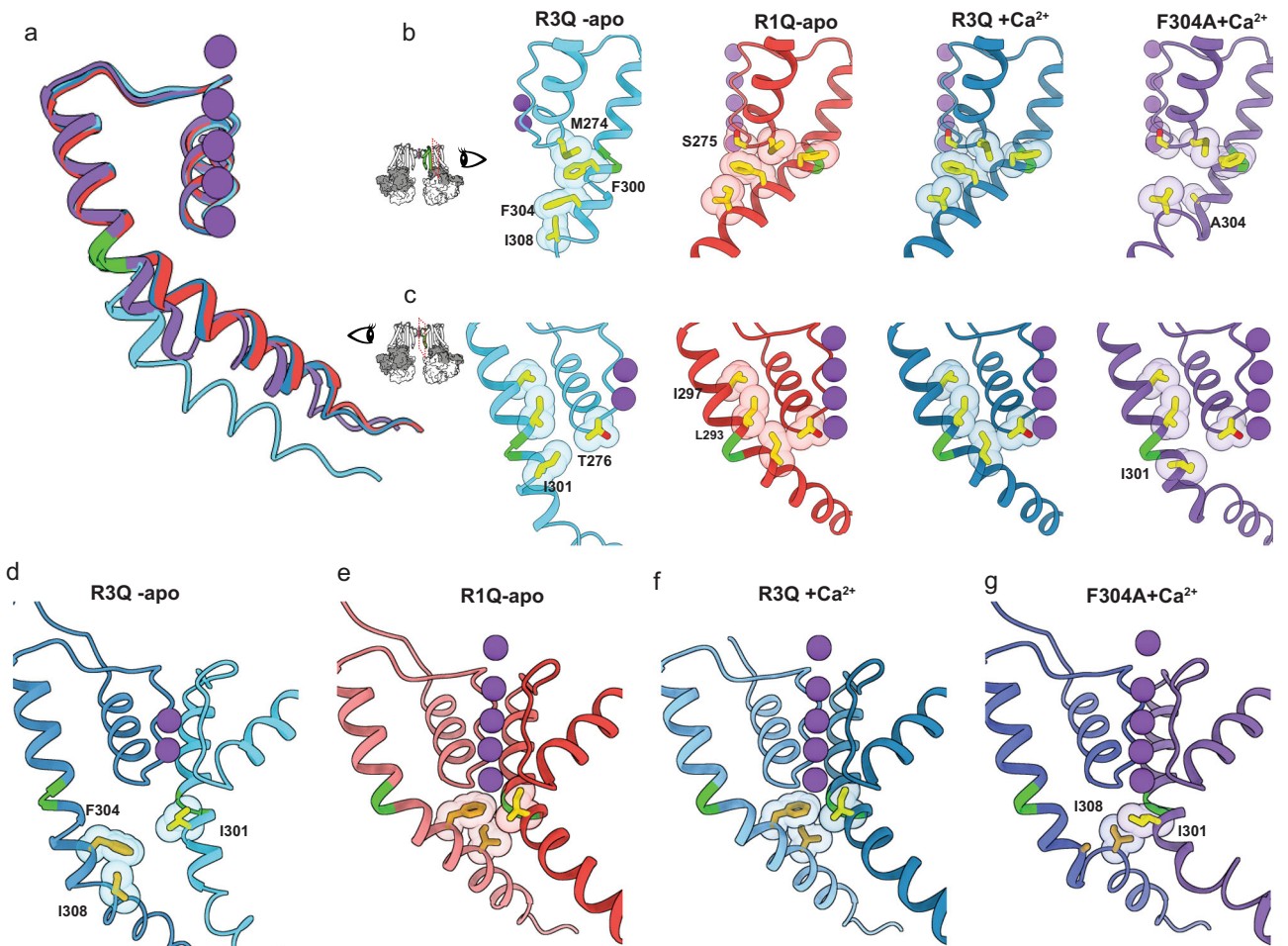

**Fig. 5 | Intersubunit coupling in the open BK channel. a** Structural comparison of pore-lining segment S6. The point at which the S6 bends is colored green. **b** Peripheral side view of the atomic model of S6 in cartoon representation with relevant hydrophobic residues in the stick representation. **c** Central side view of the atomic model of S6 in cartoon representation with relevant hydrophobic residues in the stick representation. **d**–**g** composed view of two adjacent α-monomers, highlighting the interaction between hydrophobic residues in the internal cavity.

suggests that F304 and I301 not only stabilize the activation gate but must also influence $Ca^{2+}$- and voltage-dependences. To test the idea that hydrophobic interactions affect coupling, we solved the structure of the mutant F304A in a $Ca^{2+}$-bound condition. Alanine's replacement at this position in hBK (hF315) produces channels that do not open at high $Ca^{2+}$ concentration at 0 mV[27].

We found that in the presence of $Ca^{2+}$, the voltage sensor of F304A populates the resting configuration without the electrostatic constraint imposed by the R3Q mutation. Notably, the positions of the gating charges resembled those in the R3Q mutation (Fig. S14a). The gating ring exhibits an expansion of approximately 49 Å (Fig. S14b), slightly narrower than the expansion observed for R1Q-apo and R3Q $Ca^{2+}$-bound (52 Å, Fig. S8b). Additionally, the structure of F304A $Ca^{2+}$-bound reveals a cluster of residues connecting the CDT with the VSD like that observed in $Ca^{2+}$-bound R1Q-apo and R3Q (Fig. S14c). The essential interactions between residues R223-E310 and the packing of Y212 reproduce the configuration found in the open structures.

In $Ca^{2+}$-bound F304A, rather than tilting around M298-I299 (Fig. 5a) the S6 helix moves upward near I308 (Fig. 5b) while I301 remains exposed towards the inner cavity, as in R3Q-apo (Fig. 5c). In open BK structures, the S6 hydrophobic residues move away from the conduction pathway closing the membrane-face fenestrations through hydrophobic interaction with the S6 of adjacent monomer. Specifically, F304 and I308 contact I301 from the neighboring α-monomer in open structures of R1Q apo and R3Q $Ca^{2+}$-bound (Fig. 5e, f). In the

F304A $Ca^{2+}$-bound structure, I308 interacts with I301 from the neighboring monomer, but this interaction does not lead to the tilting of S6 (Fig. 5a). This suggests the S6 helix has a hinge point at I301, F304 from adjacent monomer pivot transition to the open conformation. To test this hypothesis, we superimposed adjacent α-monomers of closed and open channels (Fig. S15). Our results showed that as the S6 helix tilts and moves upward, residues F304, I308, and P309 clash with I301 and A305 from the adjacent monomer (Fig. S15a, b). Similarly, in hBK, the homologous residues F315, V319, and P320 clash with L312 and A316 from the next monomer (Fig. S15c, d). Finally, overlaying the adjacent α-monomers of open R1Q-apo with F304A $Ca^{2+}$-bound resulted in clashes between residues of the inner cavity, suggesting that the S6 helix of F304A $Ca^{2+}$-bound has a structure like that of the closed R3Q-apo (Fig. S15e).

Alanine substitution F315A decreases the volume at this position from ~129 Å³ (Phenylalanine) to ~67 Å³ (Alanine), resulting in the channel being trapped in the closed state[27]. Even conservative modifications in the volume, such as F315I (isoleucine ~124 Å³), result in the same phenotype[57]. In contrast, mutations that increase the volume at position F315, such as Tyrosine (~154 Å³) or Tryptophan (~194 Å³), enhance BK channel activation without eliminating $Ca^{2+}$ sensitivity[27,57]. However, all these substitutions reduced the single-channel conductance to a similar extent[27,57]. This opens the possibility that a factor other than residue volume may control channel conductance. Our results indicate that the interaction between positions 304 and 301

from different monomers is crucial to transition into the open state. Residue volume appears to play some role in effectively coupling the voltage and calcium sensor, yet the stability of the open state is influenced by factors that remain elusive.

## Discussion

The $Ca^{2+}$ and voltage-activated potassium channel (BK) role as a signal integrator in excitable cells has evolved due to the remarkable cooperativity between its voltage sensors and $Ca^{2+}$-binding RCK domains. Here, a cluster of mutant BK structures has helped us define the molecular underpinnings of BK's unique voltage-sensing mechanism. Further, we present evidence leading to an understanding of how the charged residues in S4 regulate the operational cycle of the voltage sensor.

Although a shallow voltage sensitivity and fast charge transfer kinetics have been long characterized in the BK channel, the molecular mechanism of voltage sensing and the nature of its coupling to the $Ca^{2+}$-binding domains have remained a matter of debate since its early characterization[15,58]. Indeed, neutralization of the gating charges in the VSD has led to conflicting results[35–37]. Whereas Ma et al.[35], suggested that the gating particles might be distributed in S2, S3, and S4, Carrasquel-Ursulaez et al.[36] results indicate that the gating charges must be contained within the S4 transmembrane segment (R210 and R213). The behavior of BK's gating currents has also led to two contrasting interpretations regarding the degree of coupling with $Ca^{2+}$ sensors: on the one hand, the suggestion of either little or no coupling[15,28], or in contrast, a $Ca^{2+}$-driven modulation of the BK channel voltage dependence[26,36]. BK structures in the apo and $Ca^{2+}$-bound structures[5–7,29] appear consistent with strong coupling between VSD and $Ca^{2+}$ sensors.

Neutralizing these gating charges stabilizes the active and resting states of voltage sensors. We successfully determined five high-resolution structures representing BK conducting states, illustrating the independent mechanisms of voltage and calcium in channel activation. In addition, we resolved two channel structures in the closed state. These structures reveal that channel closing requires the rotation of residue I308 side chains in S6 to form a hydrophobic inner cavity, open membrane-facing fenestrations, and narrow the conduction pathway. These structures, as well as functional and computational analyses contribute to a mechanistic understanding of the BK voltage dependence. They provide a reasonable explanation for how the energy in the transmembrane electric field is transduced into pore opening, solving the debate on the nature of the closed state of the BK channel.

### The operational cycle of the voltage sensor

No significant vertical displacement was observed in S4 when resting and active structures were compared. However, we found that the VSD transitions from the resting state to the active state mostly involve reorienting S4 gating charge side chain rotamers according to the expected direction of the electric field (Fig. 2). R2 and R3 undergo rotamer reorientation with a displacement of about 5 Å in their charged moieties (Fig. 2a). However, the side chain of R196 exhibited minimal changes between the resting and active conformations of the VSD, confirming that the first arginine residue does not participate as a gating charge[36]. The charge movement is centered around the hydrophobic gasket at the VSD. In the resting state structure, the electrostatic interaction between R2 and D175 maintains the connection between the S4 and S3 segments. In contrast, in the active state, R2 becomes exposed to the extracellular environment and interacts with D142, coupling the S2 segment. Meanwhile, R3, initially positioned in the intracellular space in the resting state, moved upward to occupy the hydrophobic center. This movement is effectively an electrostatic equivalent of the 'one-click' mechanism observed in phosphatases[59,60] and KAT1 channels[61], channels in which the voltage sensor is coupled

with a cytoplasmic structural element. Yet, except for a 10° rotation around M204 α-C at the base of S4, the rotameric reorientation of the gating charges takes place with minimal displacement of the S4 helix. This suggests a distinct mechanism of voltage dependence, where the S4 helix does not behave as a translating permion, but the net charge is translocated due to the rotamer reorientation of the active gating charges.

Our findings indicate that most of the distributed charged residues in S3 and S2 exhibit state-independent interactions, which serve to hold together the various segments of the VSD and facilitate the structural rearrangements that transmit VSD energy to the pore. This is particularly true for residues D175, R156, and E169 (hD180, hR167, and hE186), which form distinct electrostatic interactions in open and closed conformations (Fig. S5b). Therefore, neutralizing these structural elements would result in a lack of coupling, as has been shown previously[35,36]. Our results support that R2 (hR210) is the main gating charge. These residues displace outward from the hydrophobic center, where the electric field drops, and couple with segment S2, resulting in conformational changes at the cytoplasmic end of the VSD connected to the pore. The degree of conformational changes exhibited by BK's VSD restricts their functional comparison with other ion channels. While Kv channels can displace a greater amount of charge (13 e⁻)[62,63], their VSD kinetics is slower than the rapid transient movement of gating charge in BK channels[58].

Our findings also reveal that the S0 helix is part of the voltage sensor. We observed an interaction between the extracellular ends of S0 (F29 and L20) with S4 (W192) (Fig. S7b). The mutation of homologous residues in S0 of hBK (F25W and L26W) by tryptophan resulted in a significant shift in the G-V relationships, affecting the voltage and calcium sensitivity[64]. Additionally, fluorometric optically tracked protein rearrangements by site-specific labeling of the S0 helix showed that it exhibited voltage dependence, similar to S4. Mutational analysis has demonstrated that hW203 (aW192) moves relative to hR20 (aR14) during VSD activation[65]. The S0 acts as a pivot for the voltage-dependent movement of S4, facilitating channel activation. This hypothesis is supported by the interaction between S0(R14) and S5 (F255) (Fig. S7b).

### Coupled rearrangement among voltage and $Ca^{2+}$ sensors

Our present structural evidence supports the hypothesis that the activation energy transmitted by voltage and calcium follows the same conformational pathway to bias pore opening. It involves conformational changes in individual sensor molecules that affect each other through allosteric interactions (Fig. 6a). These interactions result in the energy in the transmembrane electric field being transduced through small sequential changes within the transmembrane segments, specifically S0' to S4 to S5 to S6 (Fig. 6b, c). Our findings align with previous functional characterizations, showing that the degree of voltage sensor activation modulates the apparent affinity for calcium[22], and that the internal calcium concentration influences the activation energy and mid-point of activation in the VSD[26]. Furthermore, even transient conformational changes in an individual sensor appear to be coupled, as suggested through methods such as fluorometry, which tracks changes in the S4 region in response to transient pulses of calcium[25], or the movement in the CTD following transient voltage sensor activation at negative voltage[23].

We also report the opening of the channel by the exclusive activation of the calcium sensor, as predicted by allosteric models. However, we forced this scenario by locking the resting state of the VSD. In mutant R2Q, we expected to find a configuration in which the VSDs are active, but the channel is closed since this mutation impairs the coupling to the pore[36,39]. The C-linker of *Aplysia* BK might have a facilitated pore closing because of a different distribution of positive residues in the C-linker compared to hBK; in particular, the hK330 part of the RKK-ring in human BK is changed by glycine in aBK (Fig. S13a). The alanine

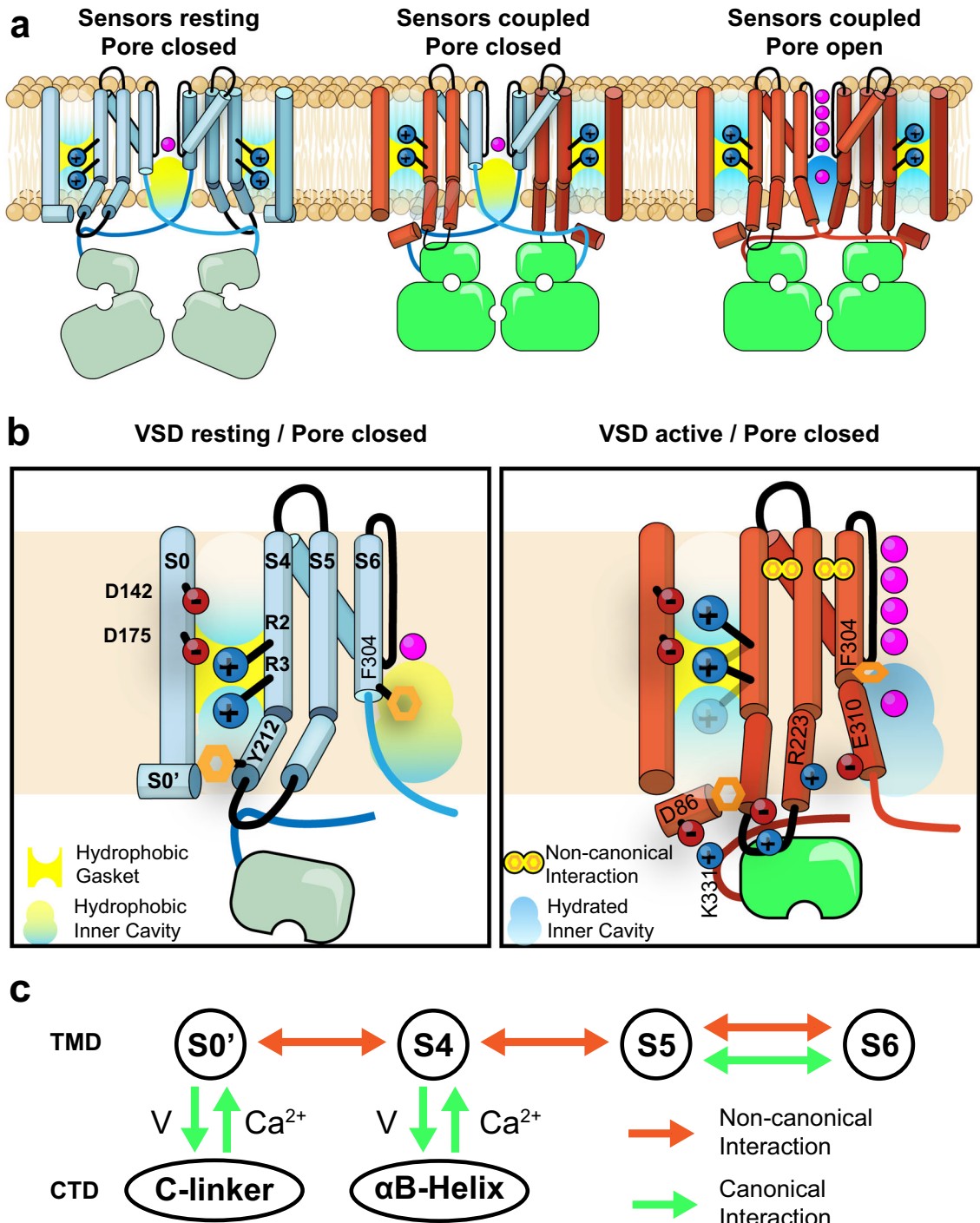

**Fig. 6 | Molecular mechanism of voltage-dependent gating in BK channel. a** the voltage-dependent channel opening involves rotary motion of the gating charges (R2 and R3) on the S4 segment, which causes the positively charged groups of R2 to cross an electric field (hydrophobic gasket). This results in the cytoplasmic end of S4 moving inward and interacting with the non-S4 helices of the VSD, facilitating electromechanical coupling. **b** The process of electromechanical coupling is facilitated by the interaction between the VSD-CTD and hydrophobic (orange) and ionic interactions participate in open pore stabilization. **c** Canonical and non-canonical interactions transfer energy from sensors via S0' → S4 → S5 → S6 in the TMD.

modification of this residue in hBK produces a 100-fold increase in the spontaneous opening of the pore[28]. Additionally, at the end of the C-linker, a cluster of charged residues ³²⁶KKER³²⁸ participates in the coupling between CTD-VSD in R1Q apo (Figs. 4c and S13a). Currently we cannot exclude the possibility that conformational changes in the VSD can exclusively open the channel. Our findings provide evidence of a complex relationship between voltage and calcium sensors and highlight a refined allosteric mechanism characterized by delicate

conformational modifications in the VSD, leading to substantial functional shifts in the channel.

**Conformational changes in the pore**

Comparing the R3Q closed structures with the wild type (WT) in its apo state shows that minor alterations in the deep pore S6 helix, particularly the R3Q mutation, significantly impact the pore size. This change leads to the exposure of hydrophobic residues (F304 and I308) along

the S6 helix into the ion conduction pathway, constricting pore diameter. The narrowest constriction occurs at I308 (radius 4.2 A). Molecular simulations have predicted a progressive de-wetting of the hydrophobic cavity when its diameter reaches less than 10 Angstroms[40]. This mechanism was previously postulated in human[44] and *Aplysia* BK[43] channels and our study provides structural evidence supporting this activation-deactivation mechanism. Notably, an artificial peptide resembling the inactivation ball of Shaker Kv acts as an open-channel blocker[66], suggesting a change in the hydrophobic nature of the closed pore. The geometric constraints imposed by the blocker bbTBA (with a diameter of 10 Å), lend credence to the notion that the R3Q-apo structure represents a tightly closed pore. bbTBA can block BK in a semi-closed state, where at least one voltage sensor is active. However, fully closed BK channels are insensitive to bbTBA blockage[34]. Recently, another pathway for the entry of hydrophobic ions and drugs was suggested for MthK[42], in which hydrophobic blockers reach the pore trough's membrane-facing fenestration. These fenestrations are present in several BK channel structures (including those reported here), where lipid or detergent density appears to partition.

The structural determination of F304A Ca$^{2+}$-bound allows for a thorough evaluation of the molecular determinants involved in opening the channel, which remains to be determined. A double glycine motif in the S6 helix of hBK has been proposed as the point of flexibility when the helix bends. This motif is not conserved in other species, yet the channel still opens at the same point in structures from different species. A state-dependent interaction between hydrophobic residues at the inner cavity seems to be relevant to the pore opening. Changes in volume at position 304 impair the transition of the S6 helix, even when the gating ring is expanded. Also, the side-fenestration at the inner cavity of F304A Ca$^{2+}$-bound is not fully closed, suggesting that the cavity still experiences a hydrophobic environment.

# Methods

### Ethical statement
Oocytes were harvested from Xenopus laevis in accordance with experimental protocols #71475 approved by the University of Chicago Institutional Animal Care and Use Committee (IACUC).

### Molecular biology and biochemistry
aSlo1, used in previous structural studies, was generously provided by Roderick Mackinnon. Residues 1 to 1070 of the *A. californica* Slo1 channel were cloned into the pFastbac vector containing a C-terminal 3 C protease site, eGFP. P0 baculovirus was generated using the Bac-to-Bac method (Invitrogen) using Cellfectin II as the transfection reagent (Thermo Fisher, 10362100). The QuikChange site-directed mutagenesis method (Agilent) was used to introduce the mutations R196Q (R1), R199Q (R2), R202Q (R3), and F304A into pFastbac-aSlo1 using KOD DNA polymerase (71085, EMD Millipore). All mutant and wild-type constructs were confirmed by DNA sequencing before structural and electrophysiological experiments.

The P0 virus was amplified once to yield the P1 baculovirus, which was used to infect sf9 cells (Thermo Fisher, 12659017) at a 3% (v/v) ratio. The cells were cultured at 27 °C in sf-900 II SFM medium (Thermo Fisher, 10902096) for 48 h before harvesting. Cells were pelleted, washed with PBS, rapidly frozen in liquid nitrogen, and stored at −80 °C until use. For purification, all steps were performed at 4 °C, and frozen cell pellets were thawed, diluted, and detergent-extracted in 20 mM Tris-HCl (pH 8.0, 320 mM KCl, 1% DDM (Anatrace D310), and 0.2% CHS (cholesteryl hemisuccinate, Sigma C6512) for 90 min. The buffer was supplemented with 2 mM EDTA (buffer A) for apo conditions, whereas for Ca$^{2+}$-bound conditions, 10 mM CaCl$_2$ and 10 mM MgCl$_2$ were added (buffer B). The solubilized supernatant was isolated by ultracentrifugation and incubated for 2 h with 1 ml CNBR-activated Sepharose beads (GE Healthcare)

coupled with 4 mg high-affinity GFP nanobodies[67]. Beads were collected by low-speed centrifugation, washed in batch mode with 10 column volumes of main buffer containing 0.2% DDM and 0.02% CHS, collected on a column by gravity, and washed with another 10 column volumes. Proteins were cleaved by the HRV 3 C protease[68] for 2–4 h, concentrated, and analyzed using SEC on a Superose 6, 10/300 GE column (GE Healthcare), with running buffer containing buffer A (or buffer B), 0.025% DDM, and 0.005% CHS. Peak fractions were collected and concentrated using a 100 kDa molecular mass cutoff centrifugal filter (Millipore concentrator unit) to 2–4 mg ml$^{-1}$. The concentrated protein was immediately used for the cryo-EM grid-freezing step.

### Cryo-EM sample preparation and imaging
Quantifoil 200-mesh 1.2/1.3 grids (Quantifoil) were plasma-cleaned for 30 s in an air mixture in a Solarus Plasma Cleaner (Gatan). Purified samples were applied onto the grids and frozen in liquid-nitrogen-cooled liquid ethane using a Vitrobot Mark IV (FEI) and the following parameters: sample volume 3.5 μl, blot time 3.5 s, blot force 3, humidity 100%, temperature 22 °C and double filter papers on each side of the Vitrobot. Grids were imaged on a Titan Krios with a K3 detector (in super-resolution mode) and GIF energy filter (set to 20 eV) at a nominal magnification of ×81,000, corresponding to a super-resolution pixel size of 0.5315 Å, 0.55 Å or 0.56 Å per pixel. The software EPU (ver. 2.13.0.1546, Thermo Fisher) was used for automated cryo-EM data collection. The videos were acquired at 1 e$^−$ A$^{-2}$ per frame for 50 frames. All the samples were acquired at The University of Chicago Advanced Electron Microscopy Core Facility (RRID: SCR_019198).

### Single-particle cryo-EM analysis
All structure determination steps were performed using Relion[69] (ver 3.1.1). All movies were binned by two and motion-corrected using Motioncor2 (ref. [70]). Contrast transfer function (CTF) estimation was performed using CTFFIND4.1 (ref. [71]). We used the SPHIRE-crYOLO package[72] for particle picking, and the coordinates were fed into the Relion for particle extraction. We picked between 100,000 and 1,500,000 initial particles for each dataset, which were processed for 2D classification. Depending on the dataset, approximately 50,000–500,000 particles were selected from the good classes. All particles were then processed for 3D refinement with C4 symmetry. Classification of the particles with C1 and C2 symmetry resulted in a map that closely resembled the overall architecture of the C4-symmetry-imposed map, albeit with a lower resolution. Post-processing of the focused TM map was performed using the star file of the K3 detector at 300 kV, and a masked nominal resolution (Table S1) was calculated according to the gold-standard 0.143 Fourier shell correlation criterion[73,74]. After a subset of particles (between 100,000 and 250,000, depending on the state) was identified for the final refinement, the particles underwent per-particle CTF refinement followed by Bayesian polishing. A final 3D refinement step was performed, followed by a post-processing step using a tighter mask and imposing C4 symmetry. The local resolution was calculated using ResMap[75].

### Model building
The model of aBK (PDB 5TJI) was used to build atomic models for the aBK mutants into our density maps. All the structural models were built into unsharpened maps. The initial models were built via iterative rounds of manual model building on COOT[76] (ver 0.9.6) and real space refinement in Phenix[77] (ver 1.14–3260). The final refined atomic models were obtained using interactive, flexible fitting using ISOLDE[78]. All structural analyses and figures were generated using UCSF ChimeraX[79] (ver 1.4–1.9).

## Electrophysiology: gating currents

Full-length WT aBK cDNA was cloned into the pBSTA vector[80]. aBK cDNA and its mutants were sequenced to ensure accurate DNA sequencing, linearized by endonuclease EcoRI (New England Biolabs, Ipswich, MA) and cleaned up with a NucleoSpin Gel and PCR Clean-up kit (Macherey-Nagel, Bethlehem, PA). cRNA was synthesized using a T7 RNA expression kit (RNA expression kit; Ambion Invitrogen, Thermo Fisher Scientific, Waltham, MA).

The follicular membrane was digested using collagenase type 2 (Worthington Biochemical Corporation, Lakewood, NJ)—2 mg/ml and supplemented by bovine serum albumin (BSA). Following the follicular membrane digestion, oocytes were incubated in standard oocytes solution (SOS) containing, in mM: 96 NaCl, 2 KCl, 1.8 CaCl2, 1 MgCl2, 0.1 EDTA, 10 HEPES and pH set to 7.4 with NaOH. SOS was supplemented with 50 μg/ml gentamycin to avoid contamination during incubation.

Approximately 24 h after surgical removal from adult frogs, 50–100 ng cRNA in 50 nl RNAse-free water was injected into enzymatically defolliculated oocytes. The injected oocytes were kept at 18 °C in 0.5× Leibovitz's L-15 medium (HyClone) supplemented with 1% horse serum, 100 U/ml penicillin, 100 μg/ml streptomycin, and 100 μg/ml amikacin.

Macroscopic currents were recorded 7 days after RNA injection, but gating current recording required a minimum of 10 days of incubation. The gating currents were recorded only in the inside-out configuration. All internal solutions contained (mM) 110 N-methyl-D-glucamine (NMDG)-MeSO3 and 10 mM HEPES. 5 mM EGTA was used as Ca$^{2+}$ chelator for "zero Ca$^{2+}$" solutions. These solutions contained an estimated 0.8 nM, based on the presence of ~10 μM contaminant [Ca$^{2+}$][81]. The external solution (pipette) contained (mM) 110 tetraethylammonium-MeSO3, 10 mM HEPES, and 2 mM MgCl2. The pH was adjusted to 7 by adding NMDG or TEA-OH according to the component of each solution. Experiments were performed at room temperature (20–22 °C). The gating charge (Q) was obtained by 300 μS integration of the Qoff response. The voltage dependency of the gating currents (Q-V) was quantified using a Boltzmann function.

$$Q(V) = \frac{Q \max}{1 + e^{\left(\frac{-z_\delta F(V - V_{0.5})}{RT}\right)}} \tag{1}$$

where $Q_{max}$ is the maximum gating charge, $V_{0.5}$ is the voltage at which $Q(V)$ equals 0.5, $F$ is Faraday's constant, $R$ is the gas constant, $T$ is the temperature in Kelvin, and $z_\delta$ is the voltage dependency.

Borosilicate glass pipettes were pulled using a Sutter micropipette puller (P-1000, Flaming/Brown). The resistance of the capillary pipettes ranged from 0.5 MΩ to 1 MΩ. The data were acquired using an Axopatch 200 B amplifier (Molecular Devices). Both the voltage command and current output were filtered at 20 kHz with 8-pole Bessel low-pass filter (Frequency Devices, Inc.). Current signals were acquired at 250 kHz using a Digidata 1440 A interface (Molecular Devices) and pCLAMP 10 acquisition software (Molecular Devices). Leak subtraction was performed using P/4 protocol[82].

## Electrophysiology: macroscopic currents sf9 infected cells

Infected SF9 cells were cultured at 27 °C in SF-900 II SFM medium (Invitrogen) for 24 h. The culture was then transferred to room temperature (20–22 °C) for an additional 24 h before recording. 5 to 10 μL of cultured cells (1 million/mL) were transferred to the recording chamber and let them attach for 10 min. The initial external solution (bath) contained (mM) 140 KCl, 1 MgCl2, 10 CaCl2, 5 Glucose, and 10 HEPES pH 6.7. The standard internal solution (pipette) contained (mM) 140 KF, 1 MgCl2, 10 EGTA, and 10 HEPES to pH 7.2. Pipettes with low resistance, approximately 100–400 kΩ (20–60 μm), were used to attempt to measure gating currents in SF9 cells; however, the aBK channel expression in the plasma membrane was low. Only microscopic currents were visible 36 to 48 h after infection. Macroscopic K+ currents were never larger than 5 nA after gigaseal formation using low-resistance pipettes. 10 CaCl2 was removed from an external solution (bath) containing (mM): 140 KCl, 1 MgCl2, 15 Glucose, and 10 HEPES pH 6.7. The voltage dependency of the macroscopic tail currents (G-V) was quantified using a Boltzmann function.

$$\frac{I}{I_{\max}} = \frac{1}{1 + e^{\left(\frac{-z_\delta F(V - V_{0.5})}{RT}\right)}} \tag{2}$$

where $I_{max}$ is the maximum current, $V_{0.5}$ is the voltage at which $I/I_{max}$ equals 0.5, $F$ is Faraday's constant, $R$ is the gas constant, $T$ is the temperature in Kelvin, and $z_\delta$ is the voltage dependency. Unless otherwise stated, chemicals were purchased from Sigma-Aldrich (St. Louis, MO).

## System construction and molecular dynamics simulations

The deposited tetramer model was used for MD simulations. This model was trimmed to include the TM segments S5 and S6 (residues 218–319) and then embedded into a 1-palmitoyl-2-oleoyl-sn-glycero-3-phosphocholine (POPC) lipid bilayer, solvated with a 150 mM KCl salt solution. The symmetry axis of the protein was aligned along the z-axis. Two K+ ions were placed at the ion binding sites of the selectivity filter: S2 and S4 of the selectivity filter, separated by two additional water molecules occupying the binding sites S1 and S3[83].Titratable residues were in their default protonation state. The final system is in an electrically neutral state, with orthorhombic periodic box dimensions of ~100 × 100 × 105 Å$^3$, consisting of ~100,000 atoms. Please see Table S2 for detailed system setup.

First, the all-atom system was energy minimized for 5000 steps, followed by a 50 ns equilibration simulation with gradually decreasing harmonic restraints applied to the protein and the K+ ions and oxygen atoms of water in the selectivity filter. Then, a 100 ns production simulation was carried out with only a weak restraint (1 kcal/mol/Å$^2$) applied to the protein backbone to count the number of cavity water molecules in the context that the protein adopts the conformation as in its cryo-EM structure. The z coordinates of the alpha carbon atoms of I297 and I308 were used to define the upper and lower boundaries of the cavity, respectively. The averaged coordinates of alpha carbon atoms of I297–I301, I301–A305, and A305–I308 from the four subunits were used to define three continuous cylinder regions in the cavity for counting water numbers inside the cavity using the program VMD[84]. Time-course analysis showed that the water number in the cavity reached convergence in our simulations. A further 350 ns production simulation was performed, and all restrains removed.

All the systems were built using CHARMM-GUI,[85] and the program VMD[84] (ver 1.9.3), and all the MD simulations were performed with the program NAMD 3.0 alpha9[86]. The CHARMM36 force field[87] was used for proteins, phospholipids, and ions, and the TIP3P model[88] for water molecules, which were widely used by the community to study the thermodynamic properties of membrane proteins and water. All simulations were carried out in an NPT ensemble with periodic boundary conditions and a time step of 2 fs. The temperature was kept at 300 K using the Langevin dynamics, and the pressure was kept at 1 atm using the Nose-Hoover Langevin piston method[89,90]. The long-range electrostatic forces were calculated with the particle-mesh Ewald method[91], and the van der Waals interaction was smoothly switched off at 10–12 Å.

## Figure preparation

Structural figures were prepared with ChimeraX[79].

## Reporting summary

Further information on research design is available in the Nature Portfolio Reporting Summary linked to this article.

## Data availability

Cryo-EM density maps of aBK mutants have been deposited in the Electron Microscopy Data Bank under accession codes (EMDB): EMD-46903 (R1Q-apo), EMD-46901 (R1Q-Ca$^{2+}$-bound), EMD-46939 (R2Q-apo), EMD-46918 (R2Q-Ca$^{2+}$-bound), EMD-46963 (R3Q-apo), EMD-46956 (R3Q-Ca$^{2+}$-bound), and EMD-46961 (F304A-Ca$^{2+}$-bound). The atomic model of the aBK mutants has been deposited in the Protein Data Bank under accession code: 9DIC (R1Q-apo), 9DI8 (R1Q-Ca$^{2+}$-bound), 9DJV (R2Q-apo), 9DIT (R2Q-Ca$^{2+}$-bound), 9DKN (R3Q-apo), 9DKF (R3Q-Ca$^{2+}$-bound), and 9DKL (F304A-Ca$^{2+}$-bound). Previously published PDB used in this article for comparation purpose: 5TJI (WT-apo), 5TJ6 (WT-Ca$^{2+}$-bound), 6ND0 (hBK-apo-mem1), 8GH9 (hBK-apo-mem2), 6V5A (hBK−L380P), 6V38 (hBK-Ca$^{2+}$-bound), and 6V3G (hBK-apo). The initial and final configurations of the molecular dynamics trajectories, as well as output of HOLE program analysis are provided in Supplementary Data 1 folder. The source data underlying Fig. 1b, and Fig. 2b are provided in Source Data file. Source data are provided with this paper.

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

## Acknowledgements

We thank Joe Austin II, James Fuller, and Tera Lavoie at The University of Chicago Advanced Electron Microscopy Core Facility (RRID: SCR_019198). We also thank Bharat Reddy, Michael Clark, Guido Mellado, and Navid Bavi for their helpful advice and discussions at all project stages. This work was supported by Fondecyt grant #1230265 (R.L.), NIH grant R01GM030376 (R.L.), and NIH grant R01-GM150272 (E.P.).

## Author contributions

G.F.C., R.L. and E.P. conceived the project. G.F.C. conducted the structural, biochemical, and electrophysiological experiments and performed all data analyses. R.S. built the models and carried out the simulations. All authors contributed to the preparation of the manuscript.

## Competing interests

The authors declare no competing interests.
