## [Transparent Peer Review file · Nature Communications]

Structural Basis of Voltage-Dependent Gating in BK Channels

Corresponding Author: Professor Eduardo Perozo

Version 0:

Reviewer comments:

Reviewer #1

(Remarks to the Author)

In this manuscript, Contreras et al provide important insights into the allosteric regulation mechanism of the BK channel, which is of great physiological importance in neurons. Although the BK channel has been extensively structurally characterized, its gating mechanism is still not fully clear. The authors designed several mutants of the gating Arg residues in the VSD, and by combining with Ca²⁺ addition, they captured the BK channel in a series of mechanistically informative states and elucidated the cooperative activation of BK channel by voltage and Ca²⁺. This is an elegant work, and I recommend publication after small revisions.

Minor points:

1. There are several typos to be fixed. Such as line 142, there is an extra word "loop" before "S2-S3 loop"; line 185, "appears play" should be "appears to play"; line 192, there is an extra word "not" after "cannot".
2. In Extended Data Fig.5, the authors showed that Mg²⁺ also binds to the Ca²⁺-bound BK channel structures. Can the authors show the density evidence or other evidence to support this claim?
3. The authors proposed that three acidic residues D175, D142 and D120 interact with the gating Arg residues and participate in VSD coupling and channel activation. The roles of D175 and D142 have been proved by respective mutants. What is the effect of D120 mutation?

Reviewer #2

(Remarks to the Author)

Contreras et al. use in the presented work cryo-EM, electrophysiological recordings and MD simulations to investigate the structural and mechanistic basis of pore gating in the voltage and Ca²⁺ dependent BK channel from *Aplysia*. BK channels are of great interest from a physiological and pathophysiological point of view, as they play as signal integrators an important role in feedback loops of several electrically excitable and non-excitable cells. The function and structure of BK channels has been investigated for decades. It is known that the activity of BK channels is cooperatively modulated by membrane voltage and cytosolic calcium concentration. However, it is not yet fully understood how this cooperativity is realized at the molecular level. In addition, BK channels differ significantly from Kv channels in terms of their voltage-dependency: BK channels show a very shallow voltage-sensitivity, but are characterized by very fast charge transfer kinetics.

In this MS, the authors present and discuss data that lead to a better understanding of both aspects of gating (voltage dependence and sensor cooperativity) in BK channels.

To understand the particular properties of the voltage sensor domain, the authors individually replaced the positively charged arginines (R196 (R1), R199 (R2), R202 (R3)) within the S4 segment of the TMD with glutamine and solved the apo and Ca²⁺-bound structures using Cryo-EM. Close examination of these structures led to the following findings:

- 1) Activation of the VSD is not accompanied by major conformational changes or vertical displacements of the S4 helix. Instead, voltage-sensitivity is caused by rotameric reorientations of the arginine side chains within the electric field. The absence of large movements of the positive arginines in the electric field explains both the shallow voltage dependence and the fast charge transfer kinetics.

2) R2 and R3 are the actual gating charges, with R2 making the largest contribution. R1 stabilizes the resting state of the VSD by the positive charge of the side chain modulating the electrostatic properties of the extracellular surface of the VSD. Replacement of R1 by Q leads to a more negative surface, which favors the upward movement of R2 and R3 residues and consequently the activation of the voltage sensor.

3) The S0 helix is part of the voltage sensor.

4) Both activation of the VSD (mimicked by R1Q apo) and activation of the Calcium sensor induce common conformational changes in the intracellular interface between the bottom of the VSD and the aB helix of the CTD. This finding provides a good explanation for the reciprocal coupling between VSD and CTD known from functional experiments.

5) R3Q apo is likely equivalent to a deep closed state which is known for the WT BK channel from functional experiments with the state-dependent pore blocker bbTBA.

6) Pore closure is caused by exposure of hydrophobic residues F304 and I308 of the S6 helix into the central pore. This generates a hydrophobic constriction which facilitates de-wetting of the cavity.

In general, the data shown is of high quality and the conclusions are comprehensible. Furthermore, based on their structural data, the authors succeed in explaining a number of known but previously mechanistically not well understood properties of BK channels and mutants. However, the authors should take a little more care with the presentation of the data and the design of the figures (see below). Furthermore, although the authors summarize their findings again in the discussion, in my opinion a concluding summary of the mechanism of pore opening is missing. Such a summary can be found in Figure 6, but not in the text.

Main Comments:

1) The authors state that the R3Q apo data set resulted in two classes with C4 symmetry (line 122). Although both classes are still shown in Figure 3, in the rest of the paper it is no longer clear which class was used for the structural comparisons. The authors should clarify this and explain their decision.

2) The authors compare in the main text several times the structural details of the generated mutants with the structure of the WT channel, but then do not show these structural details for the WT channel in the corresponding figure. For example, the authors write in line 145 "These interactions are consistent not only in R1Q but also the WT Ca²⁺ bound state". In the extended Figures 2b-d, however, the comparison to the WT structure is missing. Also in line 171, the authors refer to a structural comparison between R2Q and WT, but this is not found either in Figure 2 or in the extended Figure 2b. In order to be able to follow the statements and conclusions made in the text, it is therefore useful, in my opinion, to show the structure of the WT channel with more molecular details in the corresponding figures.

3) Regarding the coupling between voltage sensor and calcium sensor, the authors state that the main difference between R3Q-apo and WT-apo is that the aB helix in the R3Q-apo structure is broken at position L376, whereas in the WT structure the aB helix extends to R382 (line 265). It is not entirely clear to me why this should be the essential difference. The comparison of the aB helices of RQ3 apo and Ca²⁺-bound in the extended Fig. 8 provides indeed the insight that in the apo structure the aB helix starts later and breaks earlier than in the Ca²⁺-bound structure (apo: L376-K381, Ca²⁺-bound: L372-R382). However, since the helix in the proline mutant hBK L390P is even longer than in the R3Q Ca²⁺-bound structure, the start of the helix seems to be more important than the length. The authors should therefore re-examine or better justify their conclusion.

4) Some of the statements made in the text are not comprehensible when looking at the figures referred to. Some of the figures are even missing. For example:

Lines 287-290: extended Fig. 9a,b does not show a superimposition of R1Q apo VSD and R3Q apo CTD. This is actually shown in the extended Fig. 10.

Lines 274-285: The authors refer to extended Fig. 8e,f. These Figures do not exist.

In addition, some of the figure references in the text are incorrect (e.g. lines 265/266: extended Figure 8c shows hBK L390P, not R3Q-apo) or the figure captions do not describe the structures shown (e.g. extended Figure 8d). Authors should check all references, figures and captions for completeness and accuracy.

5) Figure 1b shows the voltage-dependent activation of aBK WT, R1Q, R2Q and R3Q. First, the authors should show or describe in the caption the voltage protocol used for the voltage-clamp measurements. Furthermore, only the effect of the mutations on the conductivities at equilibrium is discussed in the text. However, it is clear from the current traces shown in Figure 1b that the mutations also change the activation kinetics. Is it possible to explain these changes on the basis of the structural data shown?

Furthermore, the authors should discuss why the mutation R1Q shifts the V_{1/2} value of the G/V curve by about 150 mV to the left, while the Q/V curve in Figure 2b shows a shift of 400 mV. How can this difference of 250 mV be explained?

6) Figure 6 summarizes the proposed mechanism of allosteric coupling between VSD and CTD and the steps of pore opening. However, this figure is not actually used in the main text. It would be desirable for the reader if the authors would briefly summarize the proposed activation mechanism at the end of the discussion referencing Figure 6.

Minor comments:

1) In addition to the incorrect references to figures, there are also numerous spelling mistakes and typos throughout the text, which make it very difficult to read. The authors should therefore carefully revise the text.

2) The authors describe the interaction between a backbone nitrogen and the aromatic side chain of F or Y twice as a cation-Pi interaction (lines 141 and 339). Since the nitrogen of a peptide bond is not positively charged, this term does not seem correct to me. Amide-Pi interaction would probably fit better.

Reviewer #3

(Remarks to the Author)

The manuscript titled "Structural Basis of Voltage-Dependent Gating in BK Channels" provides an insightful exploration into the allosteric communication between the pore domain, voltage sensors, and Ca²⁺ binding sites in BK channels using an integrative approach combining cryo-EM, molecular dynamics simulations, and electrophysiological measurements. The study's primary objective is to elucidate the mechanistic coupling between voltage-sensing domains (VSD) and calcium sensors in BK channels through various mutant constructs. The manuscript correctly identifies gaps and areas of controversy in the understanding of BK channel gating mechanisms. Previous models and unresolved questions in the literature are highlighted effectively, and the study's contributions to these areas are clearly justified. This research holds significant potential for the pharmaceutical industry, particularly in developing drugs targeting BK channels. It contributes to a broader understanding of neurological and muscular diseases, which could have substantial implications for treatments and therapies.

While the research's topic is intriguing, a few issues should be addressed before the paper is considered fit for publishing.

1. Can the conformational change of VSD completely open the BK channel when the calcium sensor is in the contracted conformation?
2. At what concentration can calcium ions bind to R3Q? How does the channel open after they bind together?
3. The paper is grammatically sound, although there are several minor errors that can be easily corrected. It is worth noting that there is a reference error in line 57, 233, 560, 561, and the first letter of the title of fig. 3. and extended data fig. 2. should be capitalized.
4. Some sections, particularly the results, are data-heavy and can be overwhelming. They might benefit from further summarization to focus on the most critical findings.
5. Diagrams should appear with associated notes to increase readability.
6. Is BKR202Q channel (fully closed state) sensitive to NS1619? If so, please explain how they interact with each other.
7. Currently, the strength of the vsd-pore interaction and its relevance to activation are still controversial. Please emphasize your contributions to these two aspects.

Version 1:

Reviewer comments:

Reviewer #1

(Remarks to the Author)

The authors have addressed my concerns. I recommend publication in Nature Communications.

Reviewer #3

(Remarks to the Author)

In this manuscript (ID:NCOMMS-24-20247A), Contreras et al. present a detailed investigation of the potassium BK channels, with a focus on their bidirectional coupling mechanisms involving voltage sensing domains (VSDs) and calcium sensors. Using a combination of cryo-electron microscopy (cryo-EM), molecular dynamics (MD) simulations, and electrophysiology, the paper explores how mutations in key residues within the VSD modulate channel activation and deactivation. The study resolves long-debated structural features of BK channels, including their unique shallow voltage sensitivity, fast kinetics, and the elusive fully closed state. Overall, the research is of significant importance in the field of membrane biophysics and provides a compelling framework for understanding BK channel activity.

1. In the revised manuscript, my question has been properly addressed, but my question 2 is about R3Q [R202(R3)], and the author's answer is about the details of R2Q. No macroscopic current for R3Q mutant were detected (Fig. 1b). Interestingly, R3Q channels managed to transition to an open state under Ca²⁺-bound conditions. Therefore, I would like to know at what concentration of calcium ions would cause a conformational change. The description in lines 277-288 provides a good

answer to the latter part of the question

2. Reference 35 does not seem to mention the data for R3Q (R202Q). Please explain the description of R3Q in lin234-235.

All in all, this is a good study, and I recommend publication after minor revisions.

We thank the reviewers for their constructive comments and thorough rereading of the manuscript. Below are our point by point responses (in **bold**):

Reviewer #1 (Remarks to the Author)

In this manuscript, Contreras et al provide important insights into the allosteric regulation mechanism of the BK channel, which is of great physiological importance in neurons. Although the BK channel has been extensively structurally characterized, its gating mechanism is still not fully clear. The authors designed several mutants of the gating Arg residues in the VSD, and by combining with Ca²⁺ addition, they captured the BK channel in a series of mechanistically informative states and elucidated the cooperative activation of BK channel by voltage and Ca²⁺. This is an elegant work, and I recommend publication after small revisions.

Minor points:

1. There are several typos to be fixed. Such as line 142, there is an extra word “loop” before “S2-S3 loop”; line 185, “appears play” should be “appears to play”; line 192, there is an extra word “not” after “cannot”.

We have carefully reviewed and corrected the typos.

2. In Extended Data Fig.5, the authors showed that Mg²⁺ also binds to the Ca²⁺-bound BK channel structures. Can the authors show the density evidence or other evidence to support this claim?

We include the density in the figure Extended Data Fig.5.

3. The authors proposed that three acidic residues D175, D142 and D120 interact with the gating Arg residues and participate in VSD coupling and channel activation. The roles of D175 and D142 have been proved by respective mutants. What is the effect of D120 mutation?

D133 in the human BK promotes a 40-mV rightward shift and in the WT BK interacts with R207. In agreement with the rightward shift of the Q(V), the electrostatics predicts that its neutralization should stabilize the resting configuration of the voltage sensor (see Fig. 6 in (Carrasquel-Ursulaez et al. 2022).

Reviewer #2 (Remarks to the Author).

Contreras et al. use in the presented work cryo-EM, electrophysiological recordings and MD simulations to investigate the structural and mechanistic basis of pore gating in the voltage and Ca²⁺ dependent BK channel from *Aplysia*. BK channels are of great interest from a physiological and pathophysiological point of view, as they play an important role in feedback loops of several electrically excitable and non-excitable cells. The function and structure of BK channels has been investigated for decades. It is known that the activity of BK channels is cooperatively modulated by membrane voltage and cytosolic calcium concentration. However, it is not yet fully understood how this cooperativity is realized at the molecular level. In addition, BK channels differ significantly from Kv channels in terms of their voltage-dependency: BK channels show a very shallow voltage-sensitivity, but are characterized by very fast charge transfer kinetics.

In this MS, the authors present and discuss data that lead to a better understanding of both aspects of gating (voltage dependence and sensor cooperativity) in BK channels. To understand the particular properties of the voltage sensor domain, the authors individually replaced the positively charged arginines (R196 (R1), R199 (R2), R202 (R3)) within the S4 segment of the TMD with glutamine and solved the apo and Ca²⁺-bound structures using Cryo-EM. Close examination of these structures led to the following findings:

1) Activation of the VSD is not accompanied by major conformational changes or vertical displacements of the S4 helix. Instead, voltage-sensitivity is caused by rotameric reorientations of the arginine side chains within the electric field. The absence of large movements of the positive arginines in the electric field explains both the shallow voltage dependence and the fast charge transfer kinetics.

2) R2 and R3 are the actual gating charges, with R2 making the largest contribution. R1 stabilizes the resting state of the VSD by the positive charge of the side chain modulating the electrostatic properties of the extracellular surface of the VSD. Replacement of R1 by Q leads to a more negative surface, which favors the upward movement of R2 and R3 residues and consequently the activation of the voltage sensor.

3) The S0 helix is part of the voltage sensor.

4) Both activation of the VSD (mimicked by R1Q apo) and activation of the Calcium sensor induce common conformational changes in the intracellular interface between the bottom of the VSD and the α B helix of the CTD. This finding provides a good explanation for the reciprocal coupling between VSD and CTD known from functional experiments.

5) R3Q apo is likely equivalent to a deep closed state which is known for the WT BK channel from functional experiments with the state-dependent pore blocker bbTBA.

6) Pore closure is caused by exposure of hydrophobic residues F304 and I308 of the S6 helix into the central pore. This generates a hydrophobic constriction which facilitates de-wetting of the cavity.

In general, the data shown is of high quality and the conclusions are comprehensible. Furthermore, based on their structural data, the authors succeed in explaining a number of known but previously mechanistically not well understood properties of BK channels and

mutants. However, the authors should take a little more care with the presentation of the data and the design of the figures (see below). Furthermore, although the authors summarize their findings again in the discussion, in my opinion a concluding summary of the mechanism of pore opening is missing. Such a summary can be found in Figure 6, but not in the text.

Main Comments:

1) The authors state that the R3Q apo data set resulted in two classes with C4 symmetry (line 122). Although both classes are still shown in Figure 3, in the rest of the paper it is no longer clear which class was used for the structural comparisons. The authors should clarify this and explain their decision.

We choose the class that displayed the tightest constriction in the pore region, as a best representation of the close state. We now include a better explanation of this as a supplementary information.

2) The authors compare in the main text several times the structural details of the generated mutants with the structure of the WT channel, but then do not show these structural details for the WT channel in the corresponding figure. For example, the authors write in line 145 "These interactions are consistent not only in R1Q but also the WT Ca²⁺ bound state". In the extended Figures 2b-d, however, the comparison to the WT structure is missing. Also in line 171, the authors refer to a structural comparison between R2Q and WT, but this is not found either in Figure 2 or in the extended Figure 2b. In order to be able to follow the statements and conclusions made in the text, it is therefore useful, in my opinion, to show the structure of the WT channel with more molecular details in the corresponding figures.

Agreed. The comparisons are included in the new version of the manuscript.

3) Regarding the coupling between voltage sensor and calcium sensor, the authors state that the main difference between R3Q-apo and WT-apo is that the aB helix in the R3Q-apo structure is broken at position L376, whereas in the WT structure the aB helix extends to R382 (line 265). It is not entirely clear to me why this should be the essential difference. The comparison of the aB helices of RQ3 apo and Ca²⁺-bound in the extended Fig. 8 provides indeed the insight that in the apo structure the aB helix starts later and breaks earlier than in the Ca²⁺-bound structure (apo: L376-K381, Ca²⁺-bound: L372-R382). However, since the helix in the proline mutant hBK L390P is even longer than in the R3Q Ca²⁺-bound structure, the start of the helix seems to be more important than the length. The authors should therefore re-examine or better justify their conclusion.

We agree with the reviewer comment, and we have fixed accordingly. See lines 264-274.

4) Some of the statements made in the text are not comprehensible when looking at the figures referred to. Some of the figures are even missing. For example: Lines 287-290: extended Fig.9a,b does not show a superimposition of R1Q apo VSD and R3Q apo CTD. This is actually shown in the extended Fig. 10. Lines 274-285: The authors refer to extended Fig. 8e,f. These Figures do not exist.

In addition, some of the figure references in the text are incorrect (e.g. lines 265/266: extended Figure 8c shows hBK L390P, not R3Q-apo) or the figure captions do not describe the structures shown (e.g. extended Figure 8d). Authors should check all references, figures and captions for completeness and accuracy.

Thank you for pointing out this discrepancy. We now have properly corrected this error.

5) Figure 1b shows the voltage-dependent activation of aBK WT, R1Q, R2Q and R3Q. First, the authors should show or describe in the caption the voltage protocol used for the voltage-clamp measurements. Furthermore, only the effect of the mutations on the conductivities at equilibrium is discussed in the text. However, it is clear from the current traces shown in Figure 1b that the mutations also change the activation kinetics. Is it possible to explain these changes on the basis of the structural data shown? Furthermore, the authors should discuss why the mutation R1Q shifts the $V_{1/2}$ value of the G/V curve by about 150 mV to the left, while the Q/V curve in Figure 2b shows a shift of 400 mV. How can this difference of 250 mV be explained?

Indeed. We chose to describe the effect of the mutations at equilibrium because our structural data reflect equilibrium conditions. We have now included the voltage protocol used in the revised version of the manuscript.

We have previously shown (Carrasquel-Ursulaez et al. 2022) using the hBK that the difference between the half voltages of the Q/V and G(V) curves is mainly due to the uncoupling between the voltage sensor domain (VSD) activation and the pore opening. SO this discrepancy represents an alteration in the coupling between sensor and gate and not necessarily an intrinsic change in voltage dependence. The VSD-pore opening allosteric factor decreases from 35 in the WT BK to 24 in the R1Q mutant.

6) Figure 6 summarizes the proposed mechanism of allosteric coupling between VSD and CTD and the steps of pore opening. However, this figure is not actually used in the main text. It would be desirable for the reader if the authors would briefly summarize the proposed activation mechanism at the end of the discussion referencing Figure 6.

Thank you for catching this up. Figure 6 is now properly referenced in the main text.

Minor comments:

1) In addition to the incorrect references to figures, there are also numerous spelling mistakes and typos throughout the text, which make it very difficult to read. The authors should therefore carefully revise the text.

We appreciate the comments and have carefully reviewed and corrected the typos.

2) The authors describe the interaction between a backbone nitrogen and the aromatic side chain of F or Y twice as a cation- π interaction (lines 141 and 339). Since the nitrogen of a peptide bond is not positively charged, this term does not seem correct to me. Amide- π interaction would probably fit better.

Agreed. We have change the term, since the interaction between an aromatic ring and the backbone amine is not typically considered a cation- π interaction.

Reviewer #3 (Remarks to the Author):

The manuscript titled "Structural Basis of Voltage-Dependent Gating in BK Channels" provides an insightful exploration into the allosteric communication between the pore domain, voltage sensors, and Ca²⁺ binding sites in BK channels using an integrative approach combining cryo-EM, molecular dynamics simulations, and electrophysiological measurements. The study's primary objective is to elucidate the mechanistic coupling between voltage-sensing domains (VSD) and calcium sensors in BK channels through various mutant constructs. The manuscript correctly identifies gaps and areas of controversy in the understanding of BK channel gating mechanisms. Previous models and unresolved questions in the literature are highlighted effectively, and the study's contributions to these areas are clearly justified.

This research holds significant potential for the pharmaceutical industry, particularly in developing drugs targeting BK channels. It contributes to a broader understanding of neurological and muscular diseases, which could have substantial implications for treatments and therapies.

While the research's topic is intriguing, a few issues should be addressed before the paper is considered fit for publishing.

1. Can the conformational change of VSD completely open the BK channel when the calcium sensor is in the contracted conformation?

Currently, there is no evidence indicating that voltage can open the channel when the gating ring is contracted. Conversely, the electrophysiological, fluorescence and the present data strongly suggest that the coupling between the VSD and Ca²⁺ binding is so strong that BK channel voltage activation leads to an expansion of the gating ring even in the absence of Ca²⁺. The data of (Zhang et al. 2017) indicate that voltage can open the BK channel when the cytosolic ring is deleted, albeit with a lower maximum open probability (0.22) and a significantly reduced coupling between the VSD and Pore Domain. This is now briefly commented in the discussion.

2. At what concentration can calcium ions bind to R3Q? How does the channel open after they bind together?

Even in the absence of Ca²⁺, the R2Q channel can open at large depolarizing voltages. In the absence of Ca²⁺, the hBK conductance-voltage (G(V)) curve is rightward shifted by about 200 mV (V_{1/2} ≈ 420 mV) (Carrasquel-Ursulaez et al. 2022). (Ma, Lou, and Horrigan 2006) showed that the rBK open probability increases in the 1-50 μM Ca²⁺. However, the G(V)s are rightward shifted at all Ca²⁺ concentration. This mutant's voltage and Ca²⁺ data were best fitted using the same Ca²⁺ dissociation constant tna for the wild-type channel.

3. The paper is grammatically sound, although there are several minor errors that can be easily corrected. It is worth noting that there is a reference error in line 57, 233, 560, 561, and the first letter of the title of fig. 3. and extended data fig. 2. should be capitalized.

Thank you, we have carefully reviewed and corrected the typos.

4. Some sections, particularly the results, are data-heavy and can be overwhelming. They might benefit from further summarization to focus on the most critical findings.

We have included a summary at the end of each section to highlight the critical findings, especially in the data-heavy results sections.

5. Diagrams should appear with associated notes to increase readability.

We have summarized the proposed activation mechanism at the end of the discussion, properly referencing Figure 6.

6. Is BKR202Q channel (fully closed state) sensitive to NS1619? If so, please explain how they interact with each other.

Indeed, the BKR202Q channel is sensitive to NS1619. The binding of NS1619 to the channel is state-independent (Gessner et al. 2012). NS1619 can activate BK channels that are insensitive to calcium or voltage, such as those inhibited by heme (Tang et al. 2003). Heme inhibits the channel by disrupting the communication between sensors and the pore (Horrigan, Heinemann, and Hoshi 2005), resulting in a phenotype like R3Q channels. Our structural results indicate that NS1619 binds to the RCK1 domain of the hBK channel. Functional results suggest that NS1619 acts at the interface between RCK1 and VSD, facilitating channel opening independently of the sensor's state. Thus, NS1619 directly modulates the equilibrium of the channel gate. Therefore, NS1619 would act as a channel opener in BKR202Q.

7. Currently, the strength of the vsd-pore interaction and its relevance to activation are still controversial. Please emphasize your contributions to these two aspects.

From the electrophysiological results and the fitting to the data using the allosteric model to the BK gating kinetics, we concluded that the coupling between the VSD-PD amounts to ~ -2 kcal/mol/voltage sensor activated. However, if the four-voltage sensors are activated, the free energy of coupling is -7.2 kcal/mol (Horrigan and Aldrich 2002). These results indicate a strong coupling between the VSD and PD. The present structural data give a view of how this coupling is performed. It is important to note that the allosteric interaction between the VSD and Ca^{2+} binding is essential for this channel's physiological role in damping excitatory processes.

REFERENCES:

- Carrasquel-Ursulaez, Willy, Ignacio Segura, Ignacio Díaz-Franulic, Valeria Márquez-Miranda, Felipe Echeverría, Yenisleidy Lorenzo-Ceballos, Nicolás Espinoza, et al. 2022. "Mechanism of Voltage Sensing in Ca²⁺- and Voltage-Activated K⁺ (BK) Channels." *Proceedings of the National Academy of Sciences* 119 (25): e2204620119. <https://doi.org/10.1073/pnas.2204620119>.
- Gessner, Guido, Yong-Mei Cui, Yuko Otani, Tomohiko Ohwada, Malle Soom, Toshinori Hoshi, and Stefan H. Heinemann. 2012. "Molecular Mechanism of Pharmacological Activation of BK Channels." *Proceedings of the National Academy of Sciences* 109 (9): 3552–57. <https://doi.org/10.1073/pnas.1114321109>.
- Horrigan, Frank T, and Richard W Aldrich. 2002. "Coupling between Voltage Sensor Activation, Ca²⁺ Binding and Channel Opening in Large Conductance (BK) Potassium Channels." *The Journal of General Physiology* 120 (3): 267–305.
- Horrigan, Frank T., Stefan H. Heinemann, and Toshinori Hoshi. 2005. "Heme Regulates Allosteric Activation of the Slo1 BK Channel." *The Journal of General Physiology* 126 (1): 7–21. <https://doi.org/10.1085/jgp.200509262>.
- Ma, Zhongming, Xing Jian Lou, and Frank T Horrigan. 2006. "Role of Charged Residues in the S1-S4 Voltage Sensor of BK Channels." *The Journal of General Physiology* 127 (3): 309–28. <https://doi.org/10.1085/jgp.200509421>.
- Tang, Xiang Dong, Rong Xu, Mark F. Reynolds, Maria L. Garcia, Stefan H. Heinemann, and Toshinori Hoshi. 2003. "Haem Can Bind to and Inhibit Mammalian Calcium-Dependent Slo1 BK Channels." *Nature* 425 (6957): 531–35. <https://doi.org/10.1038/nature02003>.
- Zhang, Guohui, Yanyan Geng, Yakang Jin, Jingyi Shi, Kelli McFarland, Karl L. Magleby, Lawrence Salkoff, and Jianmin Cui. 2017. "Deletion of Cytosolic Gating Ring Decreases Gate and Voltage Sensor Coupling in BK Channels." *Journal of General Physiology* 149 (3): 373–87. <https://doi.org/10.1085/jgp.201611646>.

1. In the revised manuscript, my question has been properly addressed, but my question 2 is about R3Q [R202(R3)], and the author's answer is about the details of R2Q. No macroscopic current for R3Q mutant were detected (Fig. 1b). Interestingly, R3Q channels managed to transition to an open state under Ca²⁺-bound conditions. Therefore, I would like to know at what concentration of calcium ions would cause a conformational change. The description in lines 277-288 provides a good answer to the latter part of the question

Regarding the previous response, the authors erroneously referred to R2Q instead of R3Q. Below is the corrected answer:

Even in the absence of Ca²⁺, R3Q channels can open at large depolarizing voltages. In Ca²⁺-free conditions, the G(V) curve of hBK is shifted approximately +200 mV ($V_{1/2} \approx 420$ mV), as shown by Carraquel-Ursulaez et al. (eLife, 2022). Additionally, Ma et al. (JGP, 2006) demonstrated that rBK open probability increases across the 1–50 μ M Ca²⁺ range. However, G(V) curves remain right-shifted at all tested Ca²⁺ concentrations. Voltage- and Ca²⁺-dependent gating data for this mutant were best fitted using the same Ca²⁺ dissociation constant (K_D) as for the wild-type channel.

2. Reference 35 does not seem to mention the data for R3Q (R202Q). Please explain the description of R3Q in lin234-235.

We originally equated R3Q with charge neutralization at R3. This has now been corrected. The revised sentence reads:

"Furthermore, early functional studies showed that at saturating Ca²⁺ concentrations, the P_o at 0 mV following charge neutralization at R3 was less than 10^{-2} ."